

# Random forest as a generic framework for predictive modeling of spatial and spatio-temporal variables

Tomislav Hengl[1], Madlene Nussbaum[2], Marvin N. Wright[3], Gerard B.M. Heuvelink[4] and Benedikt Gräler[5]

[1] Envirometrix Ltd., Wageningen, Gelderland, Netherlands
[2] School of Agricultural, Forest and Food Sciences HAFL, Bern University of Applied Sciences BFH, Bern, Switzerland
[3] Leibniz Institute for Prevention Research and Epidemiology—BIPS, Bremen, Germany
[4] Soil Geography and Landscape group, Wageningen Agricultural University, Wageningen, Gelderland, Netherlands
[5] 52° North Initiative for Geospatial Open Source Software GmbH, Muenster, Germany

Corresponding author
Tomislav Hengl,
tom.hengl@gmail.com,
tom.hengl@envirometrix.net

## ABSTRACT

Random forest and similar Machine Learning techniques are already used to generate spatial predictions, but spatial location of points (geography) is often ignored in the modeling process. Spatial auto-correlation, especially if still existent in the cross-validation residuals, indicates that the predictions are maybe biased, and this is suboptimal. This paper presents a random forest for spatial predictions framework (RFsp) where buffer distances from observation points are used as explanatory variables, thus incorporating geographical proximity effects into the prediction process. The RFsp framework is illustrated with examples that use textbook datasets and apply spatial and spatio-temporal prediction to numeric, binary, categorical, multivariate and spatiotemporal variables. Performance of the RFsp framework is compared with the state-of-the-art kriging techniques using fivefold cross-validation with refitting. The results show that RFsp can obtain equally accurate and unbiased predictions as different versions of kriging. Advantages of using RFsp over kriging are that it needs no rigid statistical assumptions about the distribution and stationarity of the target variable, it is more flexible towards incorporating, combining and extending covariates of different types, and it possibly yields more informative maps characterizing the prediction error. RFsp appears to be especially attractive for building multivariate spatial prediction models that can be used as ''knowledge engines'' in various geoscience fields. Some disadvantages of RFsp are the exponentially growing computational intensity with increase of calibration data and covariates and the high sensitivity of predictions to input data quality. The key to the success of the RFsp framework might be the training data quality—especially quality of spatial sampling (to minimize extrapolation problems and any type of bias in data), and quality of model validation (to ensure that accuracy is not effected by overfitting). For many data sets, especially those with lower number of points and covariates and close-to-linear relationships, model-based geostatistics can still lead to more accurate predictions than RFsp.

# INTRODUCTION

Kriging and its many variants have been used as the Best Unbiased Linear Prediction technique for spatial points since the 1960s (*Isaaks & Srivastava, 1989*; *Cressie, 1990*; *Goovaerts, 1997*). The number of published applications on kriging has steadily increased since 1980 and the technique is now used in a variety of fields, ranging from physical geography (*Oliver & Webster, 1990*), geology and soil science (*Goovaerts, 1999*; *Minasny & McBratney, 2007*), hydrology (*Skøien, Merz & Blöschl, 2005*), epidemiology (*Moore & Carpenter, 1999*; *Graham, Atkinson & Danson, 2004*), natural hazard monitoring (*Dubois, 2005*) and climatology (*Hudson & Wackernagel, 1994*; *Hartkamp et al., 1999*; *Bárdossy & Pegram, 2013*). One of the reasons why kriging has been used so widely is its accessibility to researchers, especially thanks to the makers of gslib (*Deutsch & Journel, 1998*), ESRI's Geostatistical Analyst (http://www.esri.com), ISATIS (http://www.geovariances.com) and developers of the gstat (*Pebesma, 2004*; *Bivand et al., 2008*), geoR (*Diggle & Ribeiro Jr, 2007*) and geostatsp (*Brown, 2015*) packages for R.

Since the start of the 21st century, however, there has been an increasing interest in using more computationally intensive and primarily data-driven algorithms. These techniques are also known under the name "*machine learning*", and are applicable for various data mining, pattern recognition, regression and classification problems. One of the machine learning algorithms (MLA) that has recently proven to be efficient for producing spatial predictions is the random forest algorithm, first described in *Breiman (2001)*, and available in R through several packages such as randomForest (*Liaw & Wiener, 2002*) or the computationally faster alternative ranger (*Wright & Ziegler, 2017*). Several studies (*Prasad, Iverson & Liaw, 2006*; *Hengl et al., 2015*; *Vaysse & Lagacherie, 2015*; *Nussbaum et al., 2018*) have already shown that random forest is a promising technique for spatial prediction. Random forest, however, ignores the spatial locations of the observations and hence any spatial autocorrelation in the data not accounted for by the covariates. Modeling the relationship with covariates and spatial autocorrelation jointly using machine learning techniques is relatively novel and not entirely worked out. Using northing and easting as covariates in a random forest model may not help the prediction process as it leads to linear boundaries in the resulting map (obvious artifacts) which are directly related to the configuration of the sampling plan (*Behrens et al., in press*). A more sensible and robust use of geographical space is needed.

In this paper we describe a generic framework for spatial and spatiotemporal prediction that is based on random forest and which we refer to as "*RFsp*". With this framework we aim at including information derived from the observation locations and their spatial distribution into predictive modeling. We test whether RFsp, and potentially other tree-based machine learning algorithms, can be used as a replacement for geostatistical interpolation techniques such as ordinary and regression-kriging, i.e., kriging with external

drift. We explain in detail (using standard data sets) how to extend machine learning to general spatial prediction, and compare the prediction efficiency of random forest with that of state-of-the-art kriging methods using fivefold cross-validation with refitting the model in each subset (in the case of spatiotemporal kriging without refitting).

A complete benchmarking of the prediction efficiency is documented in R code and can be obtained via the GitHub repository at https://github.com/thengl/GeoMLA. All datasets used in this paper are either part of an existing R package or can be obtained from the GitHub repository.

## METHODS AND MATERIALS

### Spatial prediction

Spatial prediction is concerned with the prediction of the occurence, quantity and/or state of geographical phenomena, usually based on training data, e.g., ground measurements or samples $y(\mathbf{s}_i), i = 1 \ldots n$, where $\mathbf{s}_i \in D$ is a spatial coordinate (e.g., easting and northing), $n$ is the number of observed locations and $D$ is the geographical domain. Spatial prediction typically results in gridded maps or, in case of space–time prediction, animated visualizations of spatiotemporal predictions.

Model-based spatial prediction algorithms commonly aim to minimize the prediction error variance $\sigma^2(\mathbf{s}_0)$ at a prediction location $\mathbf{s}_0$ under the constraint of unbiasedness (*Christensen, 2001*). Unbiasedness and prediction error variance are defined in terms of a statistical model $\mathbf{Y} = \{Y(\mathbf{s}), \ \mathbf{s} \in D\}$ of the measurements $y(\mathbf{s}_i)$. In mathematical terms, the prediction error variance:

$$\sigma^2(\mathbf{s}_0) = \mathrm{E}\left\{\left(\hat{Y}(\mathbf{s}_0) - Y(\mathbf{s}_0)\right)^2\right\} \tag{1}$$

is to be minimized while satisfying the (unbiasedness) constraint:

$$\mathrm{E}\left\{\hat{Y}(\mathbf{s}_0) - Y(\mathbf{s}_0)\right\} = 0 \tag{2}$$

where the predictor $\hat{Y}(\mathbf{s}_0)$ of $Y(\mathbf{s}_0)$ is typically taken as a function of covariates and the $Y(\mathbf{s}_i)$ which, upon substitution of the observations $y(\mathbf{s}_i)$, yields a (deterministic) prediction $\hat{y}(\mathbf{s}_0)$.

The spatial prediction process is repeated at all nodes of a grid covering $D$ (or a space–time domain in case of spatiotemporal prediction) and produces three main outputs:

1. Estimates of the model parameters (e.g., regression coefficients and variogram parameters), i.e., the **model**;
2. Predictions at new locations, i.e., a **prediction map**;
3. Estimate of uncertainty associated with the predictions, i.e., a **prediction error variance map**.

In the case of multiple linear regression (MLR), model assumptions state that at any location in $D$ the dependent variable is the sum of a linear combination of the covariates at that location and a zero-mean normally distributed residual. Thus, at the $n$ observation locations we have:

$$\mathbf{Y} = \mathbf{X}^{\mathbf{T}} \cdot \boldsymbol{\beta} + \boldsymbol{\varepsilon} \tag{3}$$

where $\mathbf{Y}$ is a vector of the target variable at the $n$ observation locations, $\mathbf{X}$ is an $n \times p$ matrix of covariates at the same locations and $\boldsymbol{\beta}$ is a vector of $p$ regression coefficients. The stochastic residual $\boldsymbol{\varepsilon}$ is assumed to be independently and identically distributed. The paired observations of the target variable and covariates ($\mathbf{y}$ and $\mathbf{X}$) are used to estimate the regression coefficients using, e.g., Ordinary Least Squares (*Kutner et al., 2004*):

$$\hat{\boldsymbol{\beta}} = \left(\mathbf{X}^{\mathbf{T}} \cdot \mathbf{X}\right)^{-1} \cdot \mathbf{X}^{\mathbf{T}} \cdot \mathbf{y} \tag{4}$$

once the coefficients are estimated, these can be used to generate a prediction at $\mathbf{s}_0$:

$$\hat{y}(\mathbf{s}_0) = \mathbf{x}_0^{\mathbf{T}} \cdot \hat{\boldsymbol{\beta}} \tag{5}$$

with associated prediction error variance:

$$\sigma^2(\mathbf{s}_0) = var\left[\varepsilon(\mathbf{s}_0)\right] \cdot \left[1 + \mathbf{x}_0^{\mathbf{T}} \cdot \left(\mathbf{X}^{\mathbf{T}} \cdot \mathbf{X}\right)^{-1} \cdot \mathbf{x}_0\right] \tag{6}$$

here, $\mathbf{x}_0$ is a vector with covariates at the prediction location and $var\left[\varepsilon(\mathbf{s}_0)\right]$ is the variance of the stochastic residual. The latter is usually estimated by the mean squared error (MSE):

$$\text{MSE} = \frac{\sum_{i=1}^{n}(y_i - \hat{y}_i)^2}{n - p}. \tag{7}$$

The prediction error variance given by Eq. (6) is smallest at prediction points where the covariate values are in the center of the covariate ('*feature*') space and increases as predictions are made further away from the center. They are particularly large in case of extrapolation in feature space (*Kutner et al., 2004*). Note that the model defined in Eq. (3) is a non-spatial model because the observation locations and spatial-autocorrelation of the dependent variable are not taken into account.

## Kriging

Kriging is a technique developed specifically to employ knowledge about spatial autocorrelation in modeling and prediction (*Matheron, 1969*; *Christensen, 2001*; *Oliver & Webster, 2014*). Most geostatistical models assume that the target variable $Y$ at some geographic location $\mathbf{s}$ can be modeled as the sum of a deterministic mean ($\mu$) and a stochastic residual ($\varepsilon$) (*Goovaerts, 1997*; *Cressie, 2015*):

$$Y(\mathbf{s}) = \mu(\mathbf{s}) + \varepsilon(\mathbf{s}). \tag{8}$$

Assuming a constant trend ($\mu(\mathbf{s}) = \mu$ for all $\mathbf{s} \in D$), the best linear unbiased prediction (BLUP) of $y(\mathbf{s}_0)$ is given by the ordinary kriging (OK) prediction (*Goovaerts, 1997*):

$$\hat{y}_{\text{OK}}(\mathbf{s}_0) = \mathbf{w}(\mathbf{s}_0)^T \cdot \mathbf{y} \tag{9}$$

where $\mathbf{w}(\mathbf{s}_0)^T$ is a vector of kriging weights $w_i(\mathbf{s}_0), i = 1, \ldots, n$ that are obtained by minimizing the expected squared prediction error under an unbiasedness condition (i.e., the weights are forced to sum to one).

The associated prediction error variance, i.e., the OK variance, is given by (*Webster & Oliver, 2001* p.183):

$$\sigma_{\text{OK}}^2(\mathbf{s}_0) = var\left[Y(\mathbf{s}_0) - \hat{Y}(\mathbf{s}_0)\right] = C(\mathbf{s}_0, \mathbf{s}_0) - \mathbf{w}(\mathbf{s}_i)^T \cdot C_0 - \varphi, \tag{10}$$

where $C_0$ is an $n$-vector of covariances between $Y(\mathbf{s}_0)$ and the $Y(\mathbf{s}_i)$ and where $\varphi$ is a *Lagrange multiplier*.

If the distribution of the target variable is not Gaussian, a transformed Gaussian approach (*Diggle & Ribeiro Jr, 2007*, §3.8) and/or generalized linear geostatistical model approach (*Brown, 2015*) is advised. For example, the Box–Cox family of transformations is often recommended for skewed data (*Diggle & Ribeiro Jr, 2007*):

$$Y_T = \begin{cases} (Y^\eta - 1)/\eta 0 & \text{if} \quad \eta \neq 0 \\ log(Y) & \text{if} \quad \eta = 0, \end{cases} \tag{11}$$

where $\eta$ is the Box–Cox transformation parameter and $Y_T$ is the transformed target variable. The prediction and prediction error variance for log-normal simple kriging ($\mu$ known and $\eta = 0$) are obtained using (*Diggle & Ribeiro Jr, 2007*, p.61):

$$\hat{y}(\mathbf{s}_0) = \exp\left[\hat{y}_T(\mathbf{s}_0) + 0.5 \cdot \sigma_T^2(\mathbf{s}_0)\right] \tag{12}$$

$$\sigma^2(\mathbf{s}_0) = \exp\left[2 \cdot \hat{y}_T(\mathbf{s}_0) + \sigma_T^2(\mathbf{s}_0)\right] \cdot \left(\exp\left[\sigma_T^2(\mathbf{s}_0)\right] - 1\right) \tag{13}$$

where $\hat{y}_T(\mathbf{s}_0)$ and $\sigma_T^2(\mathbf{s}_0)$ are the kriging prediction and the kriging variance on the transformed scale. In other cases back-transformation can be much more difficult and may require complex approximations. Alternatively, back-transformations can be achieved using a spatial stochastic simulation approach (*Diggle & Ribeiro Jr, 2007*, Section 3.10). In this approach a very large number of realizations of the transformed variable are obtained using conditional simulation, each realization is back-transformed using the inverse of the transformation function, and summary statistics (e.g., mean, variance, quantiles) of the back-transformed realizations are computed.

The advantages of kriging are (*Webster & Oliver, 2001*; *Christensen, 2001*; *Oliver & Webster, 2014*):

- it takes a comprehensive statistical model as a starting point and derives the optimal prediction for this assumed model in a theoretically sound way;
- it exploits spatial autocorrelation in the variable of interest;
- it provides a spatially explicit measure of prediction uncertainty.

A natural extension of MLR and OK is to combine the two approaches and allow that the MLR residual of Eq. (3) is spatially correlated. This boils down to "*Regression Kriging*" (RK), "*Universal Kriging*" (UK) and/or "*Kriging with External Drift*" (KED) (*Goldberger, 1962*; *Goovaerts, 1997*; *Christensen, 2001*; *Hengl, Heuvelink & Rossiter, 2007*). UK and KED implementations are available in most geostatistical software packages (e.g., geoR and gstat) and estimate the trend coefficients and interpolate the residual in an integrated way (*Goovaerts, 1997*; *Wackernagel, 2013*), while in RK the regression and kriging are done separately. The main steps of RK are:

1. Select and prepare candidate covariates, i.e., maps of environmental and other variables that are expected to be correlated with the target variable.
2. Fit a multiple linear regression model using common procedures, while avoiding collinearity and ensuring that the MLR residuals are sufficiently normal. If required

use different type of GLM (Generalized Linear Model) to account for distribution of the target variable. If covariates are strongly correlated it may be advisable to convert these first to principal components.

3. Derive regression residuals at observation locations and fit a (residual) variogram.
4. Apply the MLR model at all prediction locations.
5. Krige the MLR residuals to all prediction locations.
6. Add up the results of steps 4 and 5.
7. Apply a back-transformation if needed.

The RK algorithm has been very successful over the past decades and is still the mainstream geostatistical technique for generating spatial predictions (*Li & Heap, 2011*). However, there are several limitations of ordinary and/or regression-kriging:

1. Kriging assumes that the residuals are normally distributed. This can often be resolved with a transformation and back-transformation, but not always. Model-based geostatistics has, at the moment, only limited solutions for zero-inflated, Poisson, binomial and other distributions that cannot easily be transformed to normality.
2. Kriging assumes that the residuals are stationary, meaning that these must have a constant mean (e.g., zero), constant variance. Often, isotropy is also assumed, meaning that the spatial autocorrelation only depends on distance, but this can be relaxed by a coordinate transformation.
3. Kriging also assumes that the variogram is known without error, i.e., it ignores variogram estimation errors (*Christensen, 2001*, pages 286–287). This can be avoided by taking a Bayesian geostatistical approach, but this complicates the analysis considerably (*Diggle & Ribeiro Jr, 2007*).
4. Most versions of kriging assume that the relation between dependent and covariates is linear, although some flexibility is offered by including transformed covariates.
5. In case of numerous possibly correlated covariates, it is very tedious to find a plausible trend model (see, e.g., *Nussbaum et al. (2018)*). Interactions among covariates are often difficult to accommodate, and usually lead to an explosion of the number of model parameters.
6. Kriging can, in the end, be computationally demanding, especially if the number of observations and/or the number of prediction locations is large.

### Random forest

Random forest (RF) (*Breiman, 2001*; *Prasad, Iverson & Liaw, 2006*; *Biau & Scornet, 2016*) is an extension of bagged trees. It has been primarily used for classification problems and several benchmarking studies have proven that it is one of the best machine learning techniques currently available (*Cutler et al., 2007*; *Boulesteix et al., 2012*; *Olson et al., 2017*).

In essence, RF is a data-driven statistical method. The mathematical formulation of the method is rather simple and instead of putting emphasis on formulating a statistical model (Fig. 1), emphasis is put on iteratively training the algorithm, using techniques such as bagging, until a "*strong learner*" is produced. Predictions in RF are generated as an ensemble estimate from a number of decision trees based on bootstrap samples (bagging).
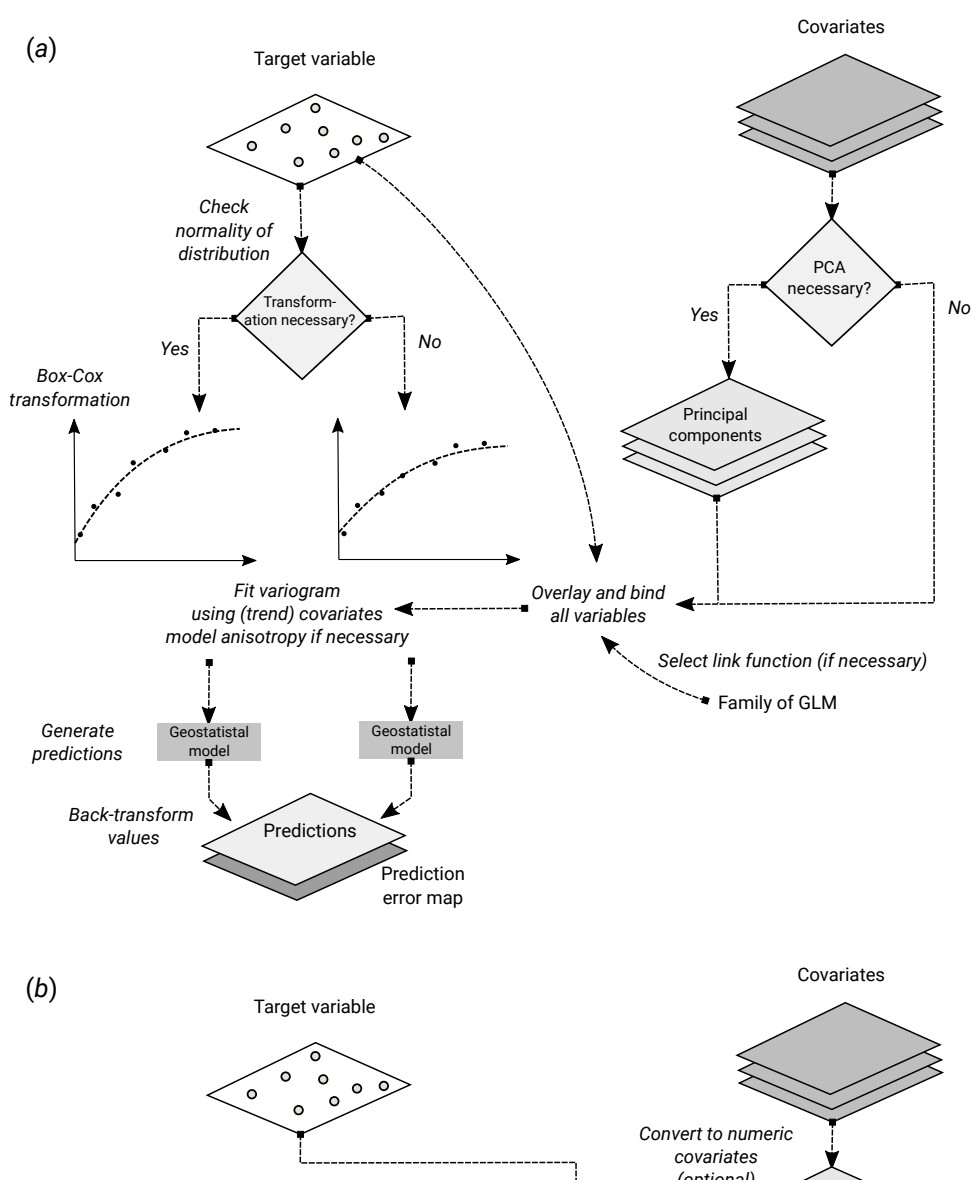

**Figure 1  Schematic difference between (A) Kriging with External Drift as implemented in the geoR package, and (B) random forest for spatial prediction.**  Being a mainly data-driven algorithm, random forest requires only limited input from the user, while model-based geostatistics requires that user specifies initial variogram parameters, anisotropy modeling, possibly transformation of the target variable and covariates and choice of a link function.

The final predictions are the average of predictions of individual trees (*Breiman, 2001*; *Prasad, Iverson & Liaw, 2006*; *Biau & Scornet, 2016*):

$$\hat{\theta}^B(x) = \frac{1}{B} \cdot \sum_{b=1}^{B} t_b^*(x), \tag{14}$$

where $b$ is the individual bootstrap sample, $B$ is the total number of trees, and $t_b^*$ is the individual learner, i.e., the individual decision tree:

$$t_b^*(x) = t(x; z_{b1}^*, \dots, z_{bK}^*), \tag{15}$$

where $z_{bk}^* (k = 1 \dots K)$ is the $k$-th training sample with pairs of values for the target variable ($y$) and covariates ($x$): $z_{bi}^* = (x_k, y_k)$.

RF, as implemented in the ranger package, has several parameters that can be fine-tuned. The most important parameters are (*Probst & Boulesteix, 2017*):

- `mtry`—number of variables to possibly split at in each node.
- `min.node.size`—minimal terminal node size.
- `sample.fraction`—fraction of observations to sample in each tree.
- `num.trees`—number of trees.

The number of trees in RF does not really need to be fine-tuned, it is recommended to set it to a computationally feasible large number (*Lopes, 2015*; *Probst & Boulesteix, 2017*).

## Uncertainty of predictions in random forest

The uncertainty of the predictions of random forest for regression-type problems can be estimated using several approaches:

- The Jackknife-after-Bootstrap method (see e.g., *Wager, Hastie & Efron (2014)*).
- The U-statistics approach of *Mentch & Hooker (2016)*.
- The Monte Carlo simulations (both target variable and covariates) approach of *Coulston et al. (2016)*.
- The Quantile Regression Forests (QRF) method (*Meinshausen, 2006*).

The approaches by *Wager, Hastie & Efron (2014)* and *Mentch & Hooker (2016)* estimate standard errors of the expected values of predictions, used to construct confidence intervals, while the approaches of *Coulston et al. (2016)* and *Meinshausen (2006)* estimate prediction intervals. Our primary interest in this article is the approach of *Meinshausen (2006)* as it can be used to produce maps of prediction error.

The Quantile Regression Forests (QRF) algorithm estimates the quantiles of the distribution of the target variable at prediction points. Thus, the 0.025 and 0.975 quantile may be used to derive the lower and upper limits of a symmetric 95% prediction interval. It does so by first deriving the random forest prediction algorithm in the usual way. While this is done with decision trees, as explained above, it ultimately boils down to a weighed linear combination of the observations:

$$\hat{y}(\mathbf{s}_0) = \sum_{i=1}^{n} \alpha_i(\mathbf{s}_0) \cdot y(\mathbf{s}_i) \tag{16}$$

in QRF, this equation is used to estimate the cumulative distribution $F_{\mathbf{s}_0}$ of $Y(\mathbf{s}_0)$, conditional to the covariates, simply by replacing the observations $y(\mathbf{s}_i)$ by an indicator transform:

$$\hat{F}_{\mathbf{s}_0}(t) = \sum_{i=1}^{n} \alpha_i(\mathbf{s}_0) \cdot 1_{y(\mathbf{s}_i) \leq t} \tag{17}$$

where $1_{y(\mathbf{s}_i) \leq t}$ is the indicator function (i.e., it is 1 if the condition is true and 0 otherwise). Any quantile $q$ of the distribution can then be derived by iterating towards the threshold $t$ for which $\hat{F}_{\mathbf{s}_0}(t) = q$. Since the entire conditional distribution can be derived in this way, it is also easy to compute the prediction error variance. For details of the algorithm, and a proof of the consistency, see *Meinshausen (2006)*.

Note that in the case of RF and QRF the prediction and associated prediction interval are derived purely using feature space and bootstrap samples. Geographical space is not included in the model as in ordinary and regression-kriging.

## Random forest for spatial data (RFsp)

RF is in essence a non-spatial approach to spatial prediction in a sense that sampling locations and general sampling pattern are ignored during the estimation of MLA model parameters. This can potentially lead to sub-optimal predictions and possibly systematic over- or under-prediction, especially where the spatial autocorrelation in the target variable is high and where point patterns show clear sampling bias. To overcome this problem we propose the following generic "*RFsp*" system:

$$Y(\mathbf{s}) = f(\mathbf{X}_G, \mathbf{X}_R, \mathbf{X}_P) \tag{18}$$

where $\mathbf{X}_G$ are covariates accounting for geographical proximity and spatial relations between observations (to mimic spatial correlation used in kriging):

$$\mathbf{X}_G = (d_{p1}, d_{p2}, \ldots, d_{pN}) \tag{19}$$

where $d_{pi}$ is the buffer distance (or any other complex proximity upslope/downslope distance, as explained in the next section) to the observed location $pi$ from $\mathbf{s}$ and $N$ is the total number of training points. $\mathbf{X}_R$ are surface reflectance covariates, i.e., usually spectral bands of remote sensing images, and $\mathbf{X}_P$ are process-based covariates. For example, the Landsat infrared band is a surface reflectance covariate, while the topographic wetness index and soil weathering index are process-based covariates. Geographic covariates are often smooth and reflect geometric composition of points, reflectance-based covariates can carry significant amount of noise and tell usually only about the surface of objects, and process-based covariates require specialized knowledge and rethinking of how to represent processes. Assuming that the RFsp is fitted only using the $\mathbf{X}_G$, the predictions would resemble OK. If all covariates are used Eq. (18), RFsp would resemble regression-kriging.

## Geographical covariates

One of the key principles of geography is that "*everything is related to everything else, but near things are more related than distant things*" (*Miller, 2004*). This principle forms the

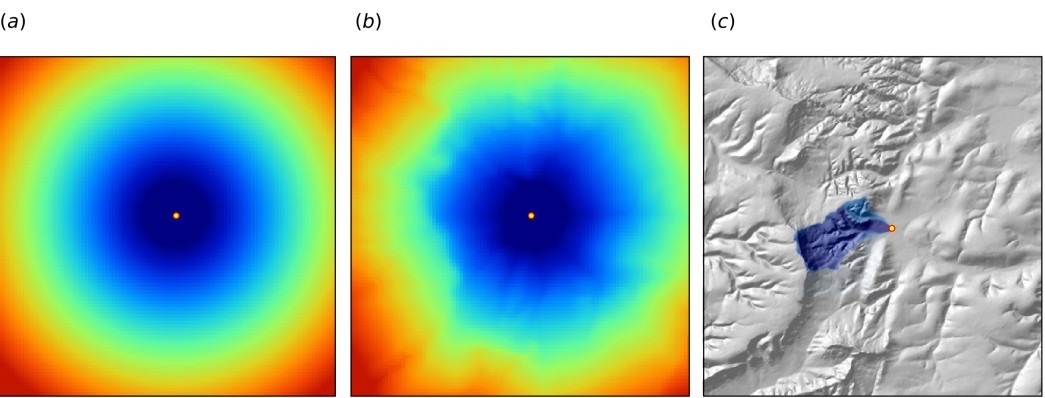

(a)        (b)        (c)

**Figure 2  Examples of distance maps to some location in space (yellow dot) based on different derivation algorithms: (A) simple Euclidean distances, (B) complex speed-based distances based on the gdistance package and Digital Elevation Model (DEM) (*Van Etten, 2017*), and (C) upslope area derived based on the DEM in SAGA GIS (*Conrad et al., 2015*).** Case study: Ebergötzen (*Böhner, McCloy & Strobl, 2006*).

basis of geostatistics, which converts this rule into a mathematical model, i.e., through spatial autocorrelation functions or variograms. The key to making RF applicable to spatial statistics problems hence lies also in preparing geographical measures of proximity and connectivity between observations, so that spatial autocorrelation is accounted for. There are multiple options for quantifying proximity and geographical connection (Fig. 2):

1. Geographical coordinates $s_1$ and $s_2$, i.e., easting and northing.
2. Euclidean distances to reference points in the study area. For example, distance to the center and edges of the study area and similar (*Behrens et al., in press*).
3. Euclidean distances to sampling locations, i.e., distances from observation locations. Here one buffer distance map can be generated per observation point or group of points. These are also distance measures used in geostatistics.
4. Downslope distances, i.e., distances within a watershed: for each sampling point one can derive upslope/downslope distances to the ridges and hydrological network and/or downslope or upslope areas (*Gruber & Peckham, 2009*). This requires, on top of using a Digital Elevation Model, a hydrological analysis of the terrain.
5. Resistance distances or weighted buffer distances, i.e., distances of the cumulative effort derived using terrain ruggedness and/or natural obstacles.

The package gdistance, for example, provides a framework to derive complex distances based on terrain complexity (*Van Etten, 2017*). Here additional input to compute complex distances are the Digital Elevation Model (DEM) and DEM-derivatives, such as slope (Fig. 2B). SAGA GIS (*Conrad et al., 2015*) offers a wide diversity of DEM derivatives that can be derived per location of interest.

In this paper we only use Eucledean buffer distances (to all sampling points) to improve RFsp predictions, but our code could be adopted to include other families of geographical covariates (as shown in Fig. 2). Note also that RF tolerates high number of covariates and multicolinearity (*Biau & Scornet, 2016*), hence multiple types of geographical covariates

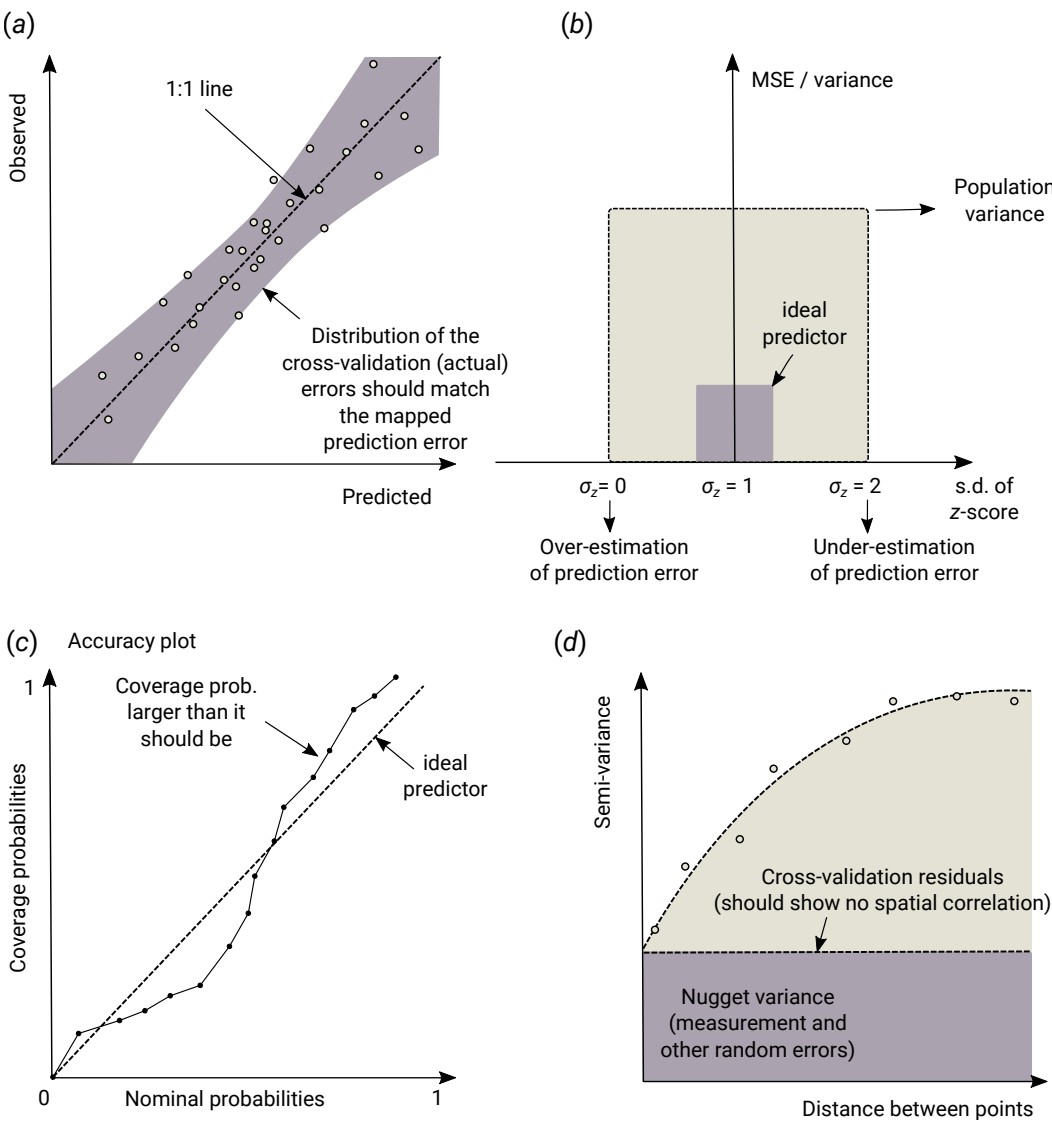

**Figure 3** Schematic examples of standard mapping performance criteria used for evaluation of spatial prediction algorithms and their interpretation: (A) predicted vs. observed plot, (B) standardized accuracy vs. standard deviation of the *z*-scores, (C) *"accuracy plots"* (after *Goovaerts (1999)*), and (D) variogram of the target variable and the cross-validation residuals. MSE, Mean Squared residual Error. In principle, all plots and statistics reported in this paper are based on the results of *n*–fold cross-validation.

(Euclidean buffer distances, upslope and downslope areas) can be used at the same time. Compare with the approach of *Behrens et al. (in press)* which only uses a combination of coordinates and the corner + center distances.

## Model performance criteria

When comparing performance of RFsp vs. OK and RK, we use the following performance criteria (Fig. 3):

1. Average RMSE based on cross-validation (CV), model R-square based on CV residuals and Concordance Correlation Coefficient—this quantifies the average accuracy of predictions i.e., amount of variation explained.
2. Average ME based on CV—this quantifies average bias in predictions.
3. Spatial autocorrelation in CV residuals—this quantifies local spatial bias in predictions.
4. Standard deviation of $z$-scores—this quantifies the reliability of estimated prediction error variances.

The RMSE and ME are derived as:

$$\text{RMSE} = \sqrt{\frac{1}{m}\sum_{j=1}^{m}(\hat{y}(\mathbf{s}_j) - y(\mathbf{s}_j))^2} \qquad (20)$$

$$\text{ME} = \frac{1}{m}\sum_{j=1}^{m}(\hat{y}(\mathbf{s}_j) - y(\mathbf{s}_j)) \qquad (21)$$

where $\hat{y}(\mathbf{s}_j)$ is the predicted value of $y$ at cross-validation location $\mathbf{s}_j$, and $m$ is the total number of cross-validation points. The amount of variation explained by the model is derived as:

$$R^2 = \left[1 - \frac{SSE}{SST}\right]\% \qquad (22)$$

where $SSE$ is the sum of squared errors at cross-validation points and $SST$ is the total sum of squares. A coefficient of determination close to 1 indicates a perfect model, i.e., 100% of variation has been explained by the model.

In addition to R-square, we also derive Lin's Concordance Correlation Coefficient (CCC) (*Steichen & Cox, 2002*):

$$\rho_c = \frac{2 \cdot \rho \cdot \sigma_{\hat{y}} \cdot \sigma_y}{\sigma_{\hat{y}}^2 + \sigma_y^2 + (\mu_{\hat{y}} - \mu_y)^2} \qquad (23)$$

where $\hat{y}$ are the predicted values and $y$ are actual values at cross-validation points, $\mu_{\hat{y}}$ and $\mu_y$ are predicted and observed means and $\rho$ is the correlation coefficient between predicted and observed values. CCC correctly quantifies how far the observed data deviate from the line of perfect concordance (1:1 line in Fig. 3A). It is usually equal to or somewhat lower than R–square, depending on the amount of bias in predictions.

The error of estimating the variance of prediction errors can likewise be quantified via the $z$-score (*Bivand et al., 2008*):

$$z_{score}(\mathbf{s}_j) = \frac{\hat{y}(\mathbf{s}_j) - y(\mathbf{s}_j)}{\sigma(\mathbf{s}_j)} \qquad (24)$$

the $z$-score are expected to have a mean equal to 0 and variance equal to 1. If the $z$-score variance is substantially smaller than 1 then the model overestimates the actual prediction uncertainty. If the $z$-score variance is substantially greater than 1 then the model underestimates the prediction uncertainty.

Note that, in the case of QRF, the method does not produce $\sigma(\mathbf{s}_j)$ but quantiles of the conditional distribution. As indicated before, the variance could be computed from the quantiles. However, since this would require computation of all quantiles at a sufficiently high discretization level, prediction error standard deviation $\sigma(\mathbf{s}_j)$ can also be estimated from the lower and upper limits of a 68.27% prediction interval:

$$\sigma_{QRF}(\mathbf{s}_j) \approx \frac{\hat{y}_{q=0.841}(\mathbf{s}_j) - \hat{y}_{q=0.159}(\mathbf{s}_j)}{2}. \tag{25}$$

This formula assumes that the prediction errors are symmetrical at each new prediction location, which might not always be the case.

## RESULTS

### Meuse data set (regression, 2D, no covariates)

In the first example, we compare the performance of a state-of-the-art model-based geostatistical model, based on the implementation in the geoR package (*Diggle & Ribeiro Jr, 2007*), with the RFsp model as implemented in the ranger package (*Wright & Ziegler, 2017*). For this we consider the Meuse data set available in the sp package:

```
> library(sp)
> demo(meuse, echo=FALSE)
```

We focus on mapping zinc (Zn) concentrations using ordinary kriging (OK) and RFsp. The assumption is that concentration of metals in soil is controlled by river flooding and carrying upstream sediments. To produce model and predictions using OK we use the package geoR. First, we fit the variogram model using the likfit function:

```
> library(geoR)

--------------------------------------------------------------
 Analysis of Geostatistical Data
 For an Introduction to geoR go to http://www.leg.ufpr.br/geoR
 geoR version 1.7-5.2 (built on 2016-05-02) is now loaded
--------------------------------------------------------------

> zinc.geo <- as.geodata(meuse["zinc"])
> ini.v <- c(var(log1p(zinc.geo$data)),500)
> zinc.vgm <- likfit(zinc.geo, lambda=0, ini=ini.v, cov.model=
"exponential")

kappa not used for the exponential correlation function
--------------------------------------------------------------
likfit: likelihood maximisation using the function optim.
likfit: Use control() to pass additional
        arguments for the maximisation function.
        For further details see documentation for optim.
likfit: It is highly advisable to run this function several
```

```
            times with different initial values for the parameters.
likfit: WARNING: This step can be time demanding!
------------------------------------------------------------
likfit: end of numerical maximisation.
```

where `lambda=0` indicates transformation by natural logarithm (positively skewed response). Once we have estimated the variogram model, we can generate predictions, i.e., the prediction map using Eq. (12):

```
> locs <- meuse.grid@coords
> zinc.ok <- krige.conv(zinc.geo, locations=locs, krige=krige.control
(obj.m=zinc.vgm))

krige.conv: model with constant mean
krige.conv: performing the Box--Cox data transformation
krige.conv: back-transforming the predicted mean and variance
krige.conv: Kriging performed using global neighbourhood
```

note here that **geoR** back-transforms the values automatically Eq. (12) preventing the user from having to find the correct unbiased back-transformation (*Diggle & Ribeiro Jr, 2007*), which is a recommended approach for less experienced users.

We compare the results of OK with **geoR** vs. RFsp. Since no other covariates are available, we use only geographical (buffer) distances to observation points. We first derive buffer distances for each individual point, using the buffer function in the **raster** package (*Hijmans & Van Etten, 2017*):

```
> grid.dist0 <- GSIF::buffer.dist(meuse["zinc"], meuse.grid[1],
as.factor
(1:nrow(meuse)))
```

which derives a gridded map for each observation point. The spatial prediction model is defined as:

```
> dn0 <- paste(names(grid.dist0), collapse="+")
> fm0 <- as.formula(paste("zinc ~ ", dn0))
```

i.e., in the formula $zinc \sim layer.1 + layer.2 + \ldots + layer.155$ which means that the target variable is a function of 155 covariates. Next, we overlay points and covariates to create a regression matrix, so that we can tune and fit a **ranger** model, and generate predictions:

```
> library(geoR)
> ov.zinc <- over(meuse["zinc"], grid.dist0)
> rm.zinc <- cbind(meuse@data["zinc"], ov.zinc)
> m.zinc <- ranger(fm0, rm.zinc, quantreg=TRUE, num.trees=150)
> m.zinc
```

```
Ranger result
Type:                              Regression
Number of trees:                   150
Sample size:                       155
Number of independent variables:   155
Mtry:                              98
Target node size:                  4
Variable importance mode:          none
OOB prediction error (MSE):        64129.11
R squared (OOB):                   0.5240641
```

```
> zinc.rfd <- predict(m.zinc, grid.dist0@data)
```

quantreg=TRUE allows to derive the lower and upper quantiles i.e. standard error of the predictions Eq. (25). The out-of-bag validation `R squared (OOB)`, indicates that the buffer distances explain about 52% of the variation in the response.

Given the different approaches, the overall pattern of the spatial predictions (maps) by OK and RFsp are surprisingly similar (Fig. 4). RFsp seems to smooth the spatial pattern more than OK, which is possibly a result of the averaging of trees in random forest. Still, overall correlation between OK and RFsp maps is high ($r = 0.97$). Compared to OK, RFsp generates a more contrasting map of standard errors with clear hotspots. Note in Fig. 4, for example, how the single isolated outlier in the lower right corner is depicted by the RFsp prediction error map. Also, note that using only coordinates as predictors results in blocky artifacts (Fig. 4C) and we do not recommended using them for mapping purposes.

The CV results show that OK is more accurate than RFsp: R-square based on fivefold cross-validation is about 0.60 (CCC = 0.76) for OK and about 0.41 (CCC = 0.55) for RFsp. Further analysis shows that in both cases there is no remaining spatial autocorrelation in the residuals (Fig. 5B). Hence, both methods have fully accounted for the spatial structure in the data. Both RFsp and OK seem to under-estimate the actual prediction error ($\sigma(z) = 1.48$ vs. $\sigma(z) = 1.28$); in this case OK yields slightly more accurate estimates of prediction error standard deviations.

Extension of RFsp with additional covariates means just adding further rasters to the buffer distances. For example, for the Meuse data set we may add global surface water occurrence (*Pekel et al., 2016*) and the LiDAR-based digital elevation model (DEM, http://ahn.nl) as potential covariates explaining zinc concentration (it is assumed that the main source of zinc in this case is the river that occasionally floods the area):

```
> meuse.grid$SWO <- readGDAL("Meuse_GlobalSurfaceWater_occurrence.
tif")$
band1[meuse.grid@grid.index]
> meuse.grid$AHN <- readGDAL("ahn.asc")$band1[meuse.grid@grid.index]
> grids.spc <- GSIF::spc(meuse.grid, as.formula("~ SWO + AHN +
ffreq + dist"))
```

```
Converting ffreq to indicators...
Converting covariates to principal components...
```

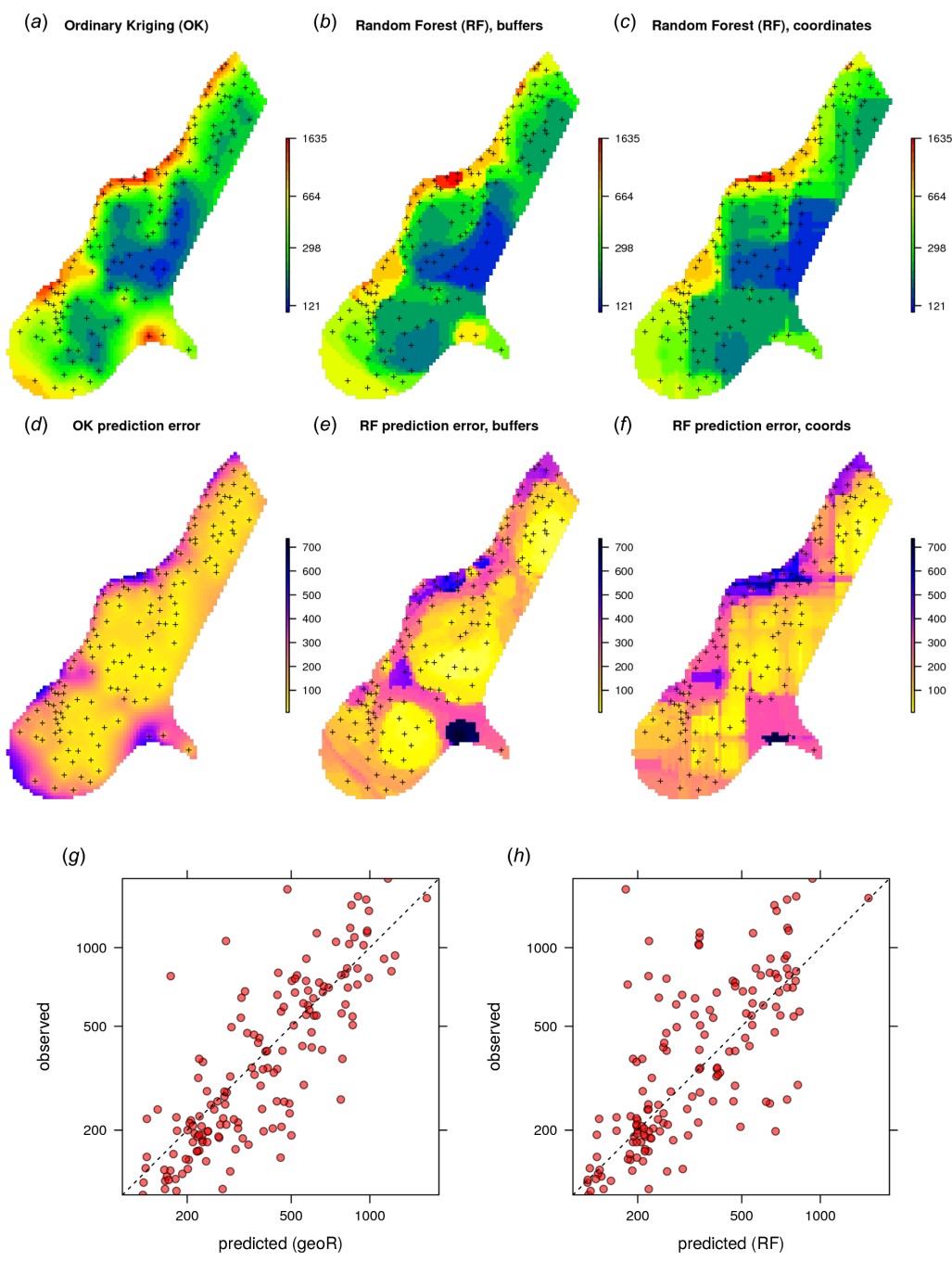

**Figure 4** Comparison of predictions based on OK as implemented in the **geoR** package (A) and random forest (B) for zinc concentrations of the Meuse dataset: predicted concentrations in log-scale (A–C), standard deviation of the prediction errors for OK and RF methods (D–F; for RF based on the **ranger** package) and correlation plots based on the fivefold cross-validation for OK and RFsp (G–H). RF with coordinates as covariates is only shown to demonstrate artifacts.

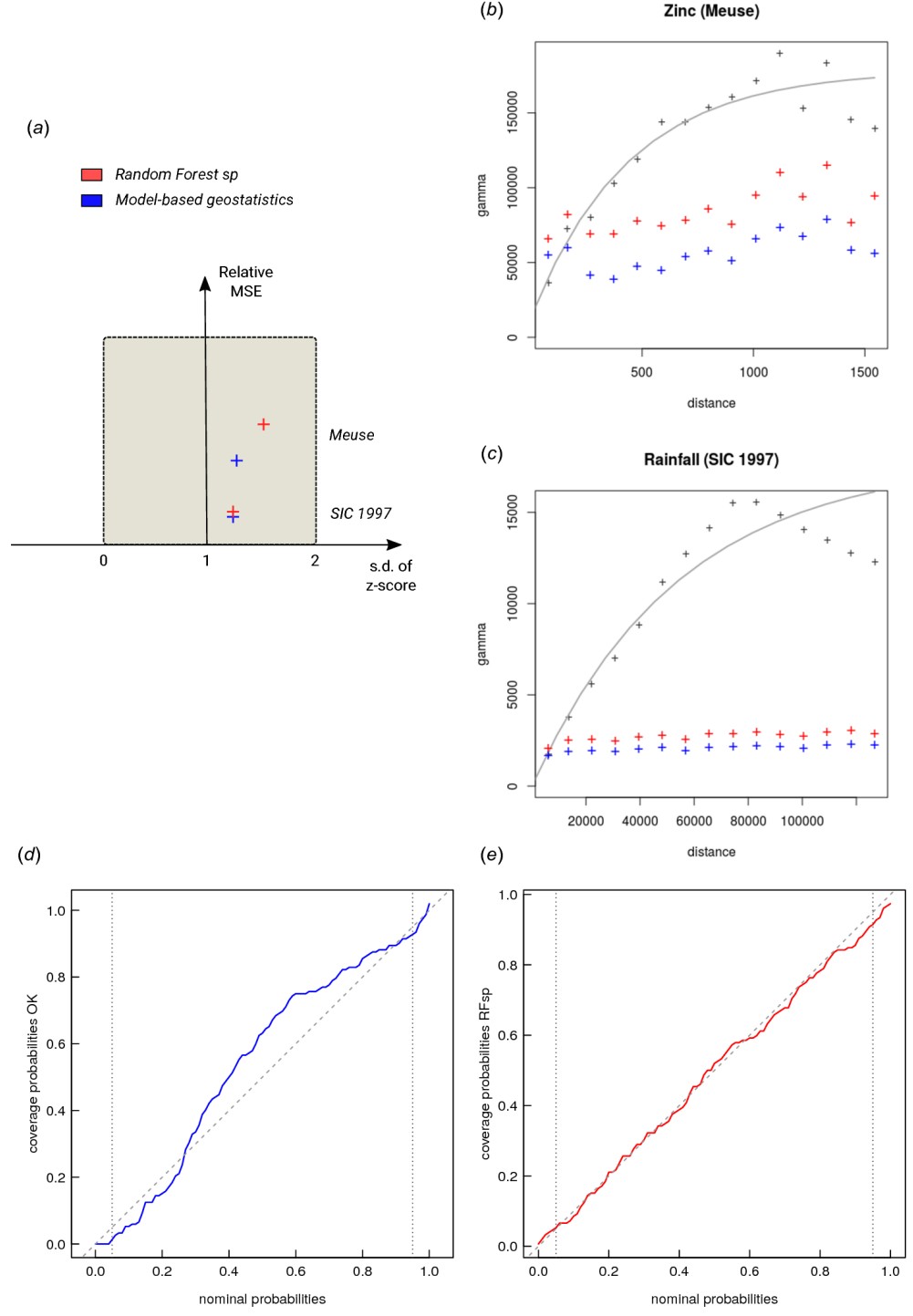

**Figure 5** Summary results of cross-validation for the Meuse (zinc) and SIC 1997 (rainfall) data sets (A) and variogram models for CV residuals (B–C). Comparison of accuracy plots for the Meuse data set (D–E). See also Fig. 3 for explanation of plots.

next, we fit the model using both thematic covariates and buffer distances:

```
> fm1 <- as.formula(paste("zinc ~ ", dn0, " + ", paste(names
(grids.spc@predicted), collapse = "+")))
> ov.zinc1 <- over(meuse["zinc"], grids.spc@predicted)
> rm.zinc1 <- cbind(meuse@data["zinc"], ov.zinc, ov.zinc1)
> m1.zinc <- ranger(fm1, rm.zinc1, mtry=130)
m1.zinc

Ranger result

Type:                               Regression
Number of trees:                    500
Sample size:                        155
Number of independent variables:    161
Mtry:                               130
Target node size:                   2
Variable importance mode:           impurity
OOB prediction error (MSE):         48124.16
R squared (OOB):                    0.6428452
```

RFsp including additional covariates results in somewhat smaller MSE than RFsp with buffer distances only. There is indeed a small difference in spatial patterns between RFsp spatial predictions derived using buffer distances only (Fig. 4) and all covariates (Fig. 6): some covariates, especially flooding frequency class and distance to the river, help with predicting zinc concentrations. Nevertheless, it seems that buffer distances are most important for mapping zinc i.e., more important than surface water occurrence, flood frequency, distance to river and elevation for producing the final predictions. This is also confirmed by the variable importance table below:

```
> xl <- as.list(ranger::importance(m1.zinc))
> print(t(data.frame(xl[order(unlist(xl), decreasing=TRUE)[1:10]])))
              [,1]
PC1        2171942.4
layer.54    835541.1
PC3         545576.9
layer.53    468480.8
PC2         428862.0
layer.118   424518.0
PC4         385037.8
layer.55    368511.7
layer.155   340373.8
layer.56    330771.0
```

which shows that, for example, points 54 and 53 are the two most influential observations, even more important than covariates (PC2–PC4) for predicting zinc concentration.

(a)
**Random Forest (RF) covs only**

(b)
**Random Forest (RF) covs + buffer dist.**

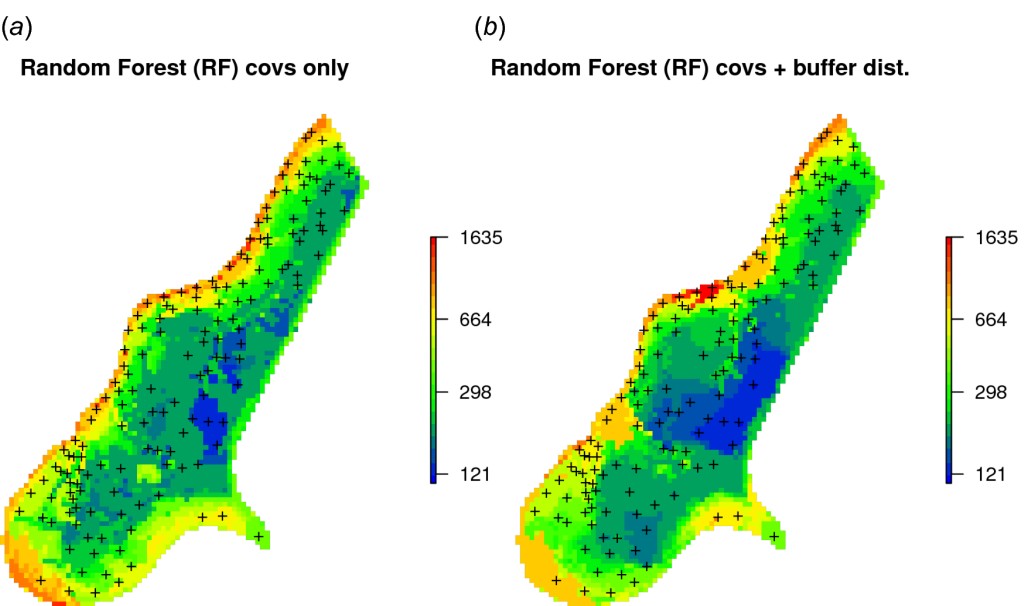

**Figure 6** **Comparison of predictions produced using random forest and covariates only (A), and random forest with covariates and buffer distances combined (B).** Compare with Fig. 4.

## Swiss rainfall dataset data set (regression, 2D, with covariates)

Another interesting dataset for comparison of RFsp with linear geostatistical modeling is the Swiss rainfall dataset used in the Spatial Interpolation Comparison (SIC 1997) exercise, described in detail in *Dubois, Malczewski & De Cort (2003)*. This dataset contains 467 measurements of daily rainfall in Switzerland on the 8th of May 1986. Possible covariates include elevation (DEM) and the long term mean monthly precipitation for May based on the CHELSA climatic images (*Karger et al., 2017*) at 1 km.

Using geoR, we can fit an RK model:

```
> sic97.sp = readRDS("./RF_vs_kriging/data/rainfall/sic97.rds")
> swiss1km = readRDS("./RF_vs_kriging/data/rainfall/swiss1km.rds")
> ov2 = over(y=swiss1km, x=sic97.sp)
> sel.d = which(!is.na(ov2$DEM))
> sic97.geo <- as.geodata(sic97.sp[sel.d,"rainfall"])
> sic97.geo$covariate = ov2[sel.d,c("CHELSA_rainfall","DEM")]
> sic.t = ~ CHELSA_rainfall + DEM
> rain.vgm <- likfit(sic97.geo, trend = sic.t, ini=c(var(log1p(sic97.geo$data))),8000),
        fix.psiA = FALSE, fix.psiR = FALSE)

------------------------------------------------------------
likfit: likelihood maximisation using the function optim.
likfit: Use control() to pass additional
        arguments for the maximisation function.
```

```
            For further details see documentation for optim.
likfit: It is highly advisable to run this function several
        times with different initial values for the parameters.
likfit: WARNING: This step can be time demanding!
----------------------------------------------------------------
likfit: end of numerical maximisation.

> rain.vgm

likfit: estimated model parameters:
      beta0         beta1         beta2         tausq        sigmasq
       phi           psiA          psiR
" 166.7679" "   0.5368" "  -0.0430" " 277.3047" "5338.1627"
"8000.0022" "   0.7796" "   5.6204"
Practical Range with cor=0.05 for asymptotic range: 23965.86

likfit: maximised log-likelihood = -2462
```

where `likfit` is the geoR function for fitting residual variograms and which produces a total of eight model coefficients: three regression coefficients (`beta`), nugget (*tausq*), sill (*sigmasq*), anisotropy ratio (*psiA*) and range (*psiR*). The rainfall data is highly anisotropic so optimizing variogram modeling through `likfit` is important (by default, geoR implements the Restricted Maximum Likelihood approach for estimation of variogram parameters, which is often considered the most reliable estimate of variogram parameters *Lark, Cullis & Welham (2006)*). The trend model:

*sic.t = ~ CHELSA_rainfall + DEM*
defines covariate variables. The final RK predictions can be generated by using the `krige.conv` function:

```
> locs2 = swiss1km@coords
> KC = krige.control(trend.d = sic.t,
    trend.l = ~ swiss1km$CHELSA_rainfall + swiss1km$DEM,
    obj.model = rain.vgm)
> rain.uk <- krige.conv(sic97.geo, locations=locs2, krige=KC)

krige.conv: model with mean defined by covariates provided by the user
krige.conv: anisotropy correction performed
krige.conv: Kriging performed using global neighbourhood
```

The results of spatial prediction using RK and RFsp are shown in Fig. 7. The cross-validation results show that in this case RFsp is nearly as accurate as RK with a cross-validation R-square of 0.78 (CCC = 0.89) vs. 0.82 (CCC = 0.91). What is striking from the Fig. 7D, however, is the high contrast of the RFsp prediction error standard deviation map, which shows a positive correlation with the values (i.e., errors are higher in areas where rainfall values are higher), but then also depicts specific areas where it seems

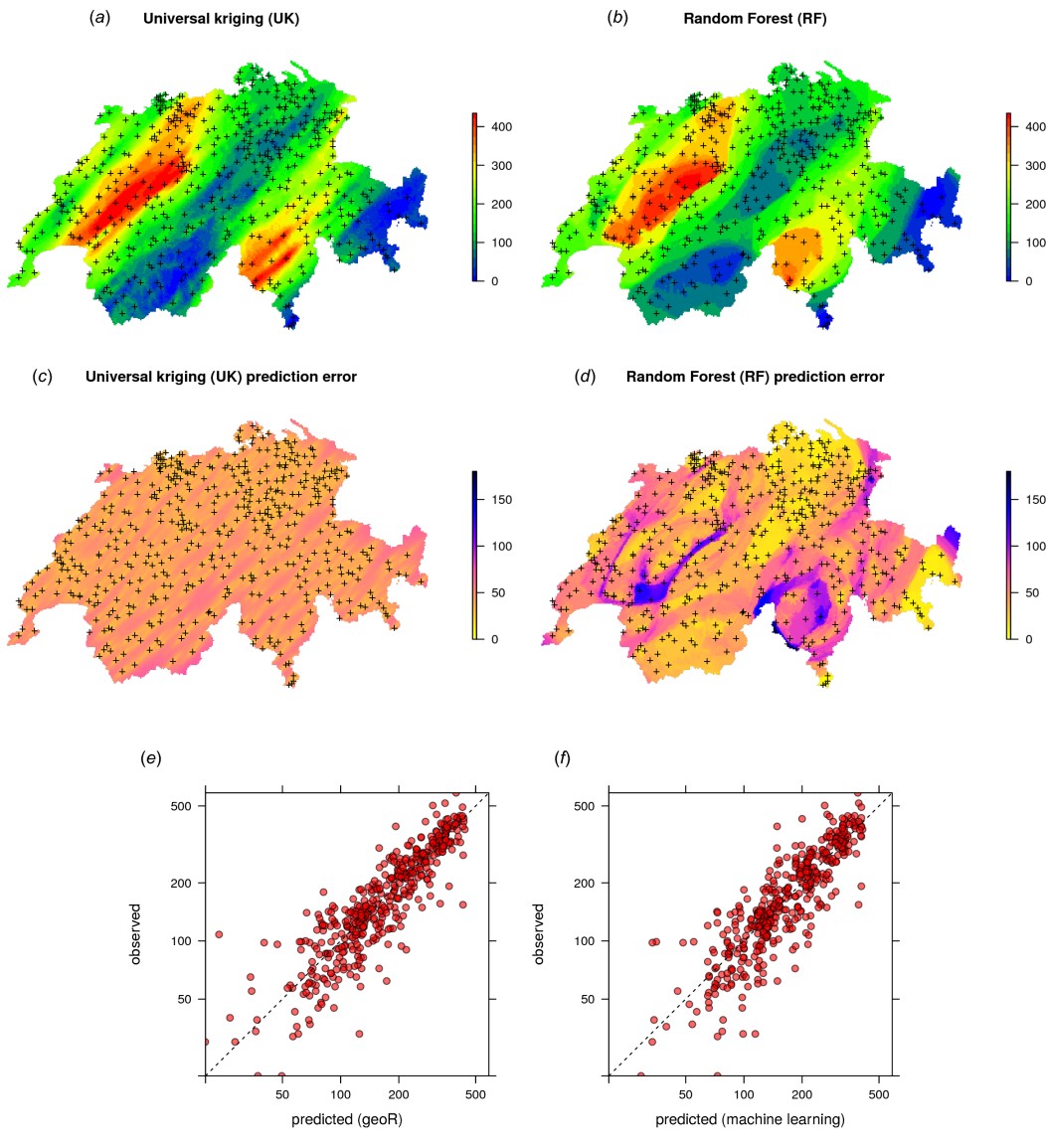

**Figure 7** **Comparison of predictions (A–B) and standard errors (C–D) produced using RK and RFsp for the Swiss rainfall data set (SIC 1997). Correlation plots for RK (E) and RFsp (F) based on fivefold cross-validation.** For more details about the dataset refer to *Dubois, Malczewski & De Cort (2003)*.

that the RF continuously produces higher prediction errors. The RK prediction error standard deviation map is much more homogeneous (Fig. 7C), mainly because of the stationarity assumption. This indicates that the RF prediction error map is potentially more informative than the UK error map. It could be used to depict local areas that are significantly more heterogeneous and complex and that require, either, denser sampling networks or covariates that better represent local processes in these areas.

The cross-validation results confirm that the prediction error standard deviations estimated by ranger and RK are both relatively similar to the actual errors. Both RFsp and RK somewhat under-estimate actual errors ($\sigma(z) = 1.16$; also visible from Figs. 7 and 5).

In this case, fitting of the variogram and generation of predictions in `geoR` takes only a few seconds, but generation of buffer distances is more computationally intensive and is in this case the bottleneck of RFsp.

## Ebergötzen data set (binomial and multinomial variables, 2D, with covariates)

As Random Forest is a generic algorithm, it can also be used to map binomial (occurrence-type) and multinomial (factor-type) responses. These are considered to be *"classification-type"* problems in Machine Learning. Mostly the same algorithms can be applied as to regression-type problems, hence the R syntax is almost the same. In traditional model-based geostatistics, factor type variables can potentially be mapped using indicator kriging (*Solow, 1986*; *Hengl et al., 2007*), but the process of fitting variograms per class and, especially for classes with few observations only, is cumbersome and unreliable.

Consider, for example, the Ebergötzen data set which contains 3,670 ground observations of soil type, and which is one of the standard datasets used in predictive soil mapping (*Böhner, McCloy & Strobl, 2006*):

```
> library(plotKML)
> data(eberg)
```

We can test predicting the probability of occurrence of soil type *"Parabraunerde"* (according to the German soil classification; Chromic Luvisols according to the World Reference Base classification) using a list of covariates and buffer distances:

```
> eberg$ParabrAunerde <- ifelse(eberg$TAXGRSC=="ParabrAunerde", "TRUE",
"FALSE")
> data(eberg_grid)
> coordinates(eberg) <- ~X+Y
> proj4string(eberg) <- CRS("+init=epsg:31467")
> gridded(eberg_grid) <- ~x+y
> proj4string(eberg_grid) <- CRS("+init=epsg:31467")
> eberg_spc <- spc(eberg_grid, ~ PRMGEO6+DEMSRT6+TWISRT6+TIRAST6)

Converting PRMGEO6 to indicators...
Converting covariates to principal components...

> eberg_grid@data <- cbind(eberg_grid@data, eberg_spc@predicted@data)
```

For `ranger`, Parabraunerde is a classification-type of problem with only two classes. We next prepare the training data by overlaying points and covariates:

```
> ov.eberg <- over(eberg, eberg_grid)
> sel <- !is.na(ov.eberg$DEMSRT6)
> eberg.dist0 <- GSIF::buffer.dist(eberg[sel,"ParabrAunerde"],
eberg_grid[2], as.factor(1:sum(sel)))
> ov.eberg2 <- over(eberg[sel,"ParabrAunerde"], eberg.dist0)
```

```
> eb.dn0 <- paste(names(eberg.dist0), collapse="+")
> eb.fm1 <- as.formula(paste("ParabrAunerde ~ ", eb.dn0, "+",
paste0("PC", 1:10, collapse = "+")))
> ov.eberg3 <- over(eberg[sel,"ParabrAunerde"], eberg_grid
[paste0("PC", 1:10)])
> rm.eberg2 <- do.call(cbind, list(eberg@data[sel,c("ParabrAunerde",
"TAXGRSC")], ov.eberg2, ov.eberg3))
```

so that predictions can be made from fitting the following model:

```
> eb.fm1
```

```
ParabrAunerde ~ layer.1 + layer.2 + layer.3 + layer.4 + layer.5 +
  ...
  layer.912 + PC1 + PC2 + PC3 + PC4 + PC5 + PC6 + PC7 + PC8 +
    PC9 + PC10
```

where `layer.*` are buffer distances to each individual point, and `PC*` are principal components based on gridded covariates. This will become a hyper-parametric model as the total number of covariates exceeds the number of observations. The fitted RF model shows:

```
> m1.ParabrAunerde <- ranger(eb.fm1, rm.eberg2[complete.cases
(rm.eberg2),],
    importance = "impurity", probability = TRUE)
> m1.ParabrAunerde
```

```
Ranger result

Type:                              Probability estimation
Number of trees:                   500
Sample size:                       829
Number of independent variables:   922
Mtry:                              30
Target node size:                  10
Variable importance mode:          impurity
OOB prediction error:              0.1536716
```

In this case the Out-of-Bag prediction error indicates a mean squared error of 0.15, which corresponds to a classification accuracy of > 85%. Note that we specify that we aim at deriving probabilities of the class of interest by setting `probability = TRUE`. The output map (Fig. 8) shows again a hybrid pattern: buffer distances to points have an effect at some locations, but this varies from area to area. Overall the most important covariates are PCs 1, 7, 8 and 3. Also note that binomial variable can be modeled with ranger as classification and/or regression-type (0/1 values) of problem—these are mathematically equivalent and should results in the same predictions i.e., predicted probabilities should matches regression predictions.

**Parabraunerde class (RF)**

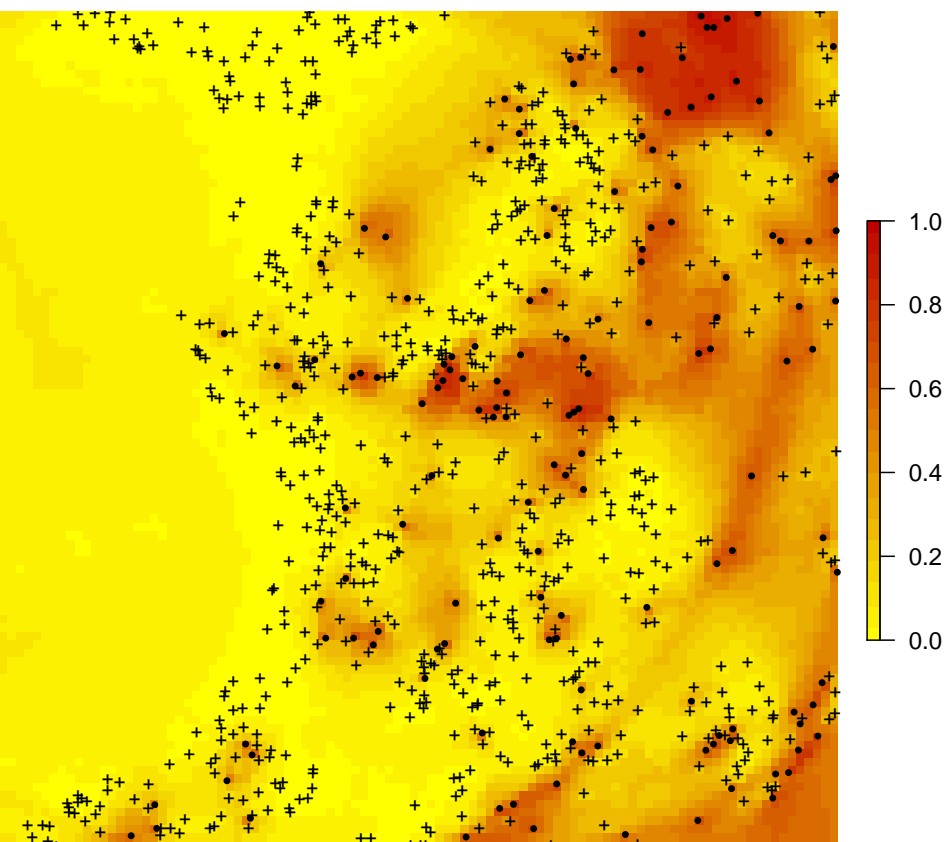

**Figure 8** **Predicted distribution for the Parabraunerde occurence probabilities (the Ebergötzen data set) produced using buffer distances combined with other covariates.** Dots indicate observed occurrence locations (TRUE) for the class, crosses indicate non-occurrence locations (FALSE). Predictions reveal a hybrid spatial pattern that reflects both geographical proximity (samples) and relationship between soil class and landscape (covariate or feature space).

In a similar way we can also map all other soil types (Fig. 9). The function `GSIF::autopredict` wraps all steps described previously into a single function:

```
> soiltype <- GSIF::autopredict(eberg["TAXGRSC"], eberg_grid,
auto.plot=FALSE)

Generating buffer distances...
Converting PRMGEO6 to indicators...
Converting LNCCOR6 to indicators...
Converting covariates to principal components...
Fitting a random forest model using 'ranger'...
Generating predictions...
```

In this case buffer distances are derived to each class, which is less computationally intensive than deriving distances to each individual observation locations because there are typically

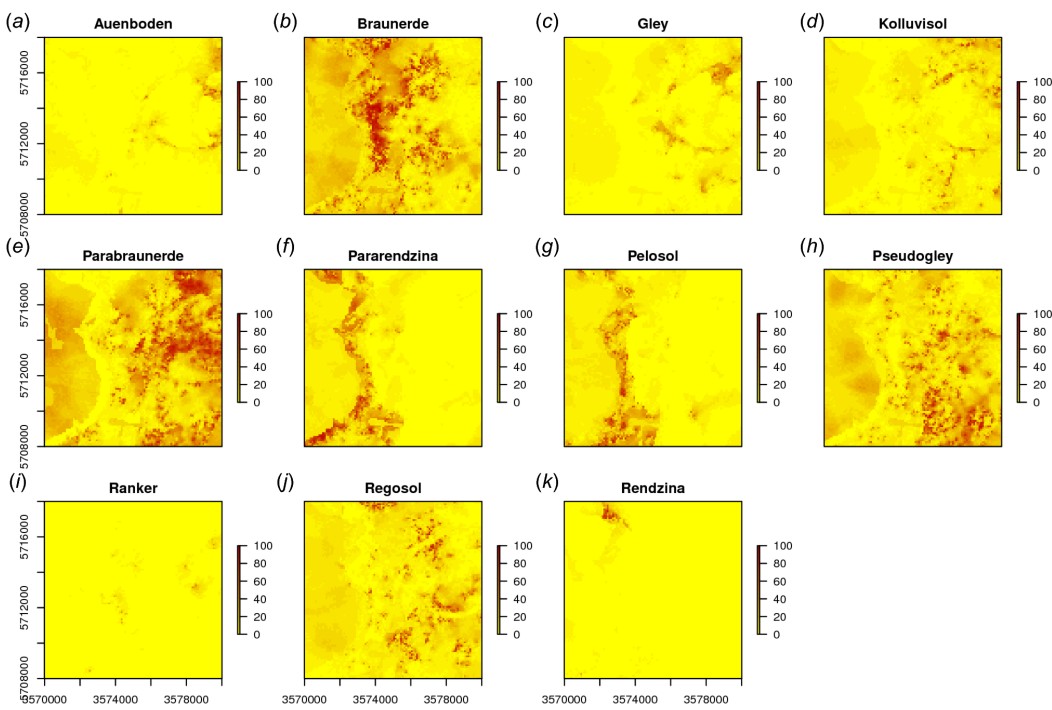

**Figure 9** **Predicted soil type occurrence probabilities (A–K) for the Ebergötzen data set (German soil classification system) using buffer distance to each class and a stack of covariates representing parent material, hydrology and land cover.**

much fewer classes than observations. Although deriving buffer distances to each individual observation location provides certainly more detail, in the case of factor-type variables, RF might benefit well from only the distances to classes.

In summary, spatial prediction of binary and factor-type variables is straightforward with ranger, and buffer distances can be incorporated in the same way as for continuous-numerical variables. In geostatistics, handling categorical dependent variables is more complex, where the GLGM with link functions and/or indicator kriging would need to be used, among others requiring that variograms are fitted per class.

## NRCS data set (weighted regression, 3D)

In many cases training data sets (points) come with variable measurement errors or have been collected with a sampling bias. If information about the data quality of each individual observation is known, then it also makes sense to use this information to produce a more balanced spatial prediction model. The package ranger allows this via the argument `case.weights`—observations with larger weights will be selected with higher probability in the bootstrap, so that the output model will be (correctly) more influenced by observations with higher weights.

Consider, for example, the soil point data set prepared as a combination of (a) the National Cooperative Soil Survey (NCSS) Characterization Database, and (b) National Soil Information System (NASIS) points (*Ramcharan et al., 2018*). The NCSS soil points

contain laboratory measurements of soil clay content, while the NASIS points contain only soil texture classes determined by hand (from which also clay content can be derived), hence with much higher measurement error:

```
> carson <- read.csv("./RF_vs_kriging/data/NRCS/carson_CLYPPT.csv")
> carson1km <- readRDS("./RF_vs_kriging/data/NRCS/carson_covs1km.rds")
> coordinates(carson) <- ~ X + Y
> proj4string(carson) = carson1km@proj4string
> carson$DEPTH.f = ifelse(is.na(carson$DEPTH), 20, carson$DEPTH)
```

The number of NASIS points is much higher (ca. 5×) than that of the NCSS points, but the NCSS observations are about 3× more accurate. We do not actually know what the exact measurement errors for each observation so we take a pragmatic approach and set the weights in the modeling procedure proportional to the quality of data:

```
> str(carson@data)

'data.frame':  3418 obs. of  8 variables:
 $ X.1     : int  1 2 3 4 5 6 8 9 10 11 ...
 $ SOURCEID: Factor w/ 3230 levels "00CA693X017jbf",..: 1392
 1393 3101 3102 ...
 $ pscs    : Factor w/ 25 levels "ASHY","ASHY OVER CLAYEY",..: 19 7 16
 16 16 16 16 7 20 20 ...
 $ CLYPPT  : int  20 64 27 27 27 27 27 64 20 20 ...
 $ CLYPPT.sd: int  8 16 6 6 6 6 6 16 8 8 ...
 $ SOURCEDB: Factor w/ 2 levels "NASIS","NCSS": 1 1 1 1 1 1 1 1 1 1 ...
 $ DEPTH   : int  NA NA NA NA NA NA NA NA NA NA ...
 $ DEPTH.f : num  20 20 20 20 20 20 20 20 20 20 ...
```

where CLYPPT is the estimated clay fraction (m%) of the fine earth, and CLYPPT.sd is the reported measurement error standard deviation associated to each individual point (in this case soil horizon). We can build a weighted RF spatial prediction model using:

```
> rm.carson <- cbind(as.data.frame(carson), over(carson["CLYPPT"],
carson1km))
> fm.clay <- as.formula(paste("CLYPPT ~ DEPTH.f + ", paste(names
(carson1km),
collapse = "+")))
> pars.carson <- list(num.trees=150, mtry=25, case.weights=1/
(rm.carson.s$CLYPPT.sd^2))
> m.clay <- ranger(fm.clay, rm.carson, unlist(pars.carson))
```

In this case we used $1/\Delta\sigma_y^2$, i.e., inverse measurement variance as `case.weights` so that points that were measured in the lab will receive much higher weights.

Figure 10B shows that, in this specific case, the model without weights seems to predict somewhat higher values, especially in the extrapolation areas. Also the prediction error

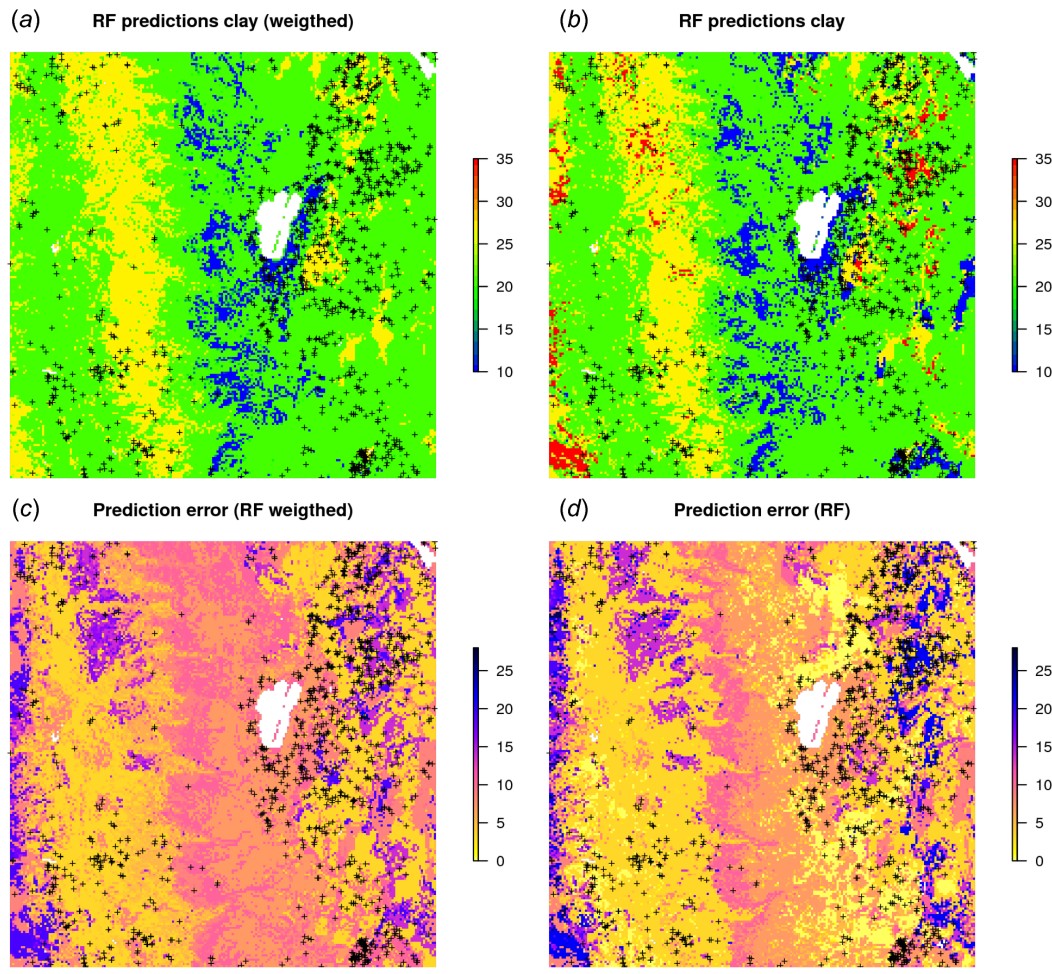

**Figure 10** RF predictions (A–B) and prediction error standard deviations (C–D) for clay content with and without using measurement errors as weights. Study area around Lake Tahoe, California, USA. Point data sources: National Cooperative Soil Survey (NCSS) Characterization Database and National Soil Information System (NASIS) (*Ramcharan et al., 2018*).

standard deviations seems to be somewhat smaller (ca. 10%) for the unweighted regression model. This indicates that using measurement errors in model calibration is important and one should not avoid specifying this in the model, especially if the training data is heterogeneous.

## The National Geochemical Survey data set, multivariate case (regression, 2D)

Because RF is a decision tree-based method, this opens a possibility to model multiple variables within a single model, i.e., by using type of variable as a covariate. This means that prediction values will show discrete jumps, depending on which variable type is used. The general form of such model is:

$$Y(\mathbf{s}) = f\left\{Y_{\text{type}}, C_{\text{type}}, \mathbf{X}_G, \mathbf{X}_R, \mathbf{X}_P\right\} \qquad (26)$$

where $Y_{type}$ is the variable type, i.e., chemical element, $C_{type}$ specifies the sampling or laboratory method used, and **X** are the covariates from Eq. (18).

Consider for example the National Geochemical Survey database that contains over 70,000 sampling points spread over the USA (*Grossman et al., 2004*). Here we use a subset of this dataset with 2,858 points with measurements of Pb, Cu, K and Mg covering the US states Illinois and Indiana. Some useful covariates to help explain the distribution of elements in stream sediments and soils have been previously prepared (*Hengl, 2009*) and include:

```
> geochem <- readRDS("./RF_vs_kriging/data/geochem/geochem.rds")
> usa5km <- readRDS("./RF_vs_kriging/data/geochem/usa5km.rds")
> str(usa5km@data)

'data.frame':  16000 obs. of  6 variables:
 $ geomap  : Factor w/ 17 levels "6","7","8","13",..: 9 9 9 9 9
 9 9 9 9 9 ...
 $ globedem: num  266 269 279 269 269 271 284 255 253 285 ...
 $ dTRI    : num  0.007 0.007 0.008 0.008 0.009 ...
 $ nlights03: num  6 5 0 5 0 1 5 13 5 5 ...
 $ dairp   : num  0.035 0.034 0.035 0.036 0.038 ...
 $ sdroads : num  0 0 5679 0 0 ...
```

where `geomap` is the geological map of the USA, `globedem` is elevation, `dTRI` is the density of industrial pollutants (based on the the pan-American Environmental Atlas of pollutants), `nlights03` is the lights at night image from 2003, `dairp` is the density of traffic based on main roads and railroads and `sdroads` is distance to main roads and railroads.

Since the task is to build a single model using a list of chemical elements, we need to combine all target variables into a single regression matrix. In R this can be achieved by using:

```
> geochem <- spTransform(geochem, CRS(proj4string(usa5km)))
> usa5km.spc <- spc(usa5km, ~geomap+globedem+dTRI+nlights03+dairp+
sdroads)

Converting geomap to indicators...
Converting covariates to principal components...

> ov.geochem <- over(x=geochem, y=usa5km.spc@predicted)
> df.lst <- lapply(c("PB_ICP40","CU_ICP40","K_ICP40","MG_ICP40"),
  function(i){cbind(geochem@data[,c(i,"TYPEDESC")],
  ov.geochem)})
```

next, we rename columns that contain the target variable:

```
> t.vars = c("PB_ICP40","CU_ICP40","K_ICP40","MG_ICP40")
> df.lst = lapply(t.vars, function(i){cbind(geochem@data[,c
(i,"TYPEDESC")], ov.geochem)})
```

```
> names(df.lst) = t.vars
> for(i in t.vars){colnames(df.lst[[i]])[1] = "Y"}
> for(i in t.vars){df.lst[[i]]$TYPE = i}
```
so that all variables (now called Y) can be combined into a single regression matrix:

```
> rm.geochem = do.call(rbind, df.lst)
> str(rm.geochem)

'data.frame':  11432 obs. of  25 variables:
 $ Y       : num  9 10 10 9 16 14 8 15 11 9 ...
 $ TYPE    : chr  "PB_ICP40" "PB_ICP40" "PB_ICP40" "PB_ICP40" ...
 ...
```
where the TYPE column carries the information of the type of variable. To this regression matrix we can fit a RF model of the shape:

```
> fm.g

Y ~ PC1 + PC2 + PC3 + PC4 + PC5 + PC6 + PC7 + PC8 + PC9 + PC10 +
    PC11 + PC12 + PC13 + PC14 + PC15 + PC16 + PC17 + PC18 + PC19 +
    PC20 + PC21 + TYPECU_ICP40 + TYPEK_ICP40 + TYPEMG_ICP40 +
    TYPEPB_ICP40 + TYPEDESCSOIL + TYPEDESCSTRM.SED.DRY +
    TYPEDESCSTRM.SED.WET + TYPEDESCUNKNOWN
```

where PC* are the principal components derived from covariates, TYPECU_ICP40 is an indicator variable defining whether the variable is Cu, TYPEK_ICP40 is an indicator variable for K, TYPEDESCSOIL is an indicator variable for soil sample (362 training points in total), and TYPEDESCSTRM.SED.WET is an indicator variable for stream sediment sample (2,233 training points in total).

The RF fitted to these data gives:

```
> rm.geochem.e <- rm.geochem.e[complete.cases(rm.geochem.e),]
> m1.geochem <- ranger(fm.g, rm.geochem.e, importance = "impurity")
> m1.geochem

Ranger result

Type:                              Regression
Number of trees:                   500
Sample size:                       11148
Number of independent variables:   29
Mtry:                              5
Target node size:                  5
Variable importance mode:          impurity
OOB prediction error (MSE):        1462.767
R squared (OOB):                   0.3975704
```

To predict values and generate maps we need to specify (a) type of chemical element, and (b) type of sampling medium at the new predictions locations:

```
> new.usa5km = usa5km.spc@predicted@data
> new.usa5km$TYPEDESCSOIL = 0
> new.usa5km$TYPEDESCSTRM.SED.DRY = 0
> new.usa5km$TYPEDESCSTRM.SED.WET = 1
> new.usa5km$TYPEDESCUNKNOWN = 0
> for(i in t.vars){
  new.usa5km[,paste0("TYPE",i)] = 1
  for(j in t.vars[!t.vars %in% i]){ new.usa5km[,paste0("TYPE",j)]
  = 0 }
  x <- predict(m1.geochem, new.usa5km)
  usa5km@data[,paste0(i,"_rf")] = x$predictions
}
```

The results of the prediction are shown in Fig. 11. From the produced maps, we can see that the spatial patterns of the four elements are relatively independent (apart from Pb and Cu which seem to be highly cross-correlated), even though they are based on a single RF model. Note that, just by switching the TYPEDES, we could produce predictions for a variety of combinations of sampling conditions and chemical elements.

A disadvantage of running multivariate models is that the data size increases rapidly and hence also the computing intensity. For a comparison, the National Geochemical Survey comprises hundreds of chemical elements hence the total size of training points could easily exceed several millions. In addition, computation of model diagnostics such as variable importance becomes difficult as all variables are included in a single model—ranger indicates an overall R-square of 0.40, but not all chemical elements can be mapped with the same accuracy. On the other hand, it appears that extension from univariate to multivariate spatial predictions models is fairly straightforward and can be compared to various co-kriging techniques used in the traditional geostatistics (*Pebesma, 2004*). Note also that an R package already exists—IntegratedMRF (*Rahman, Otridge & Pal, 2017*)—which takes multiple output responses, and which could probably be integrated with RFsp.

### Daily precipitation Boulder (CO) data set (regression, 2D+T)

In the last example we look at extending 2D regression based on RFsp to spatiotemporal data, i.e., to a 2D+T case. For this we use a time series of daily precipitation measurements obtained from https://www.ncdc.noaa.gov for the period of 2014–2017 for the area around Boulder, Colorado (available via GitHub repository). We can load the data by:

```
> co_prec = readRDS("./RF_vs_kriging/data/st_prec/boulder_prcp.rds")
> str(co_prec)

'data.frame':  176467 obs. of  16 variables:
 $ STATION : Factor w/ 239 levels "US1COBO0004",..: 64 64 64 64 64
  64 64 64 64 64 ...
```

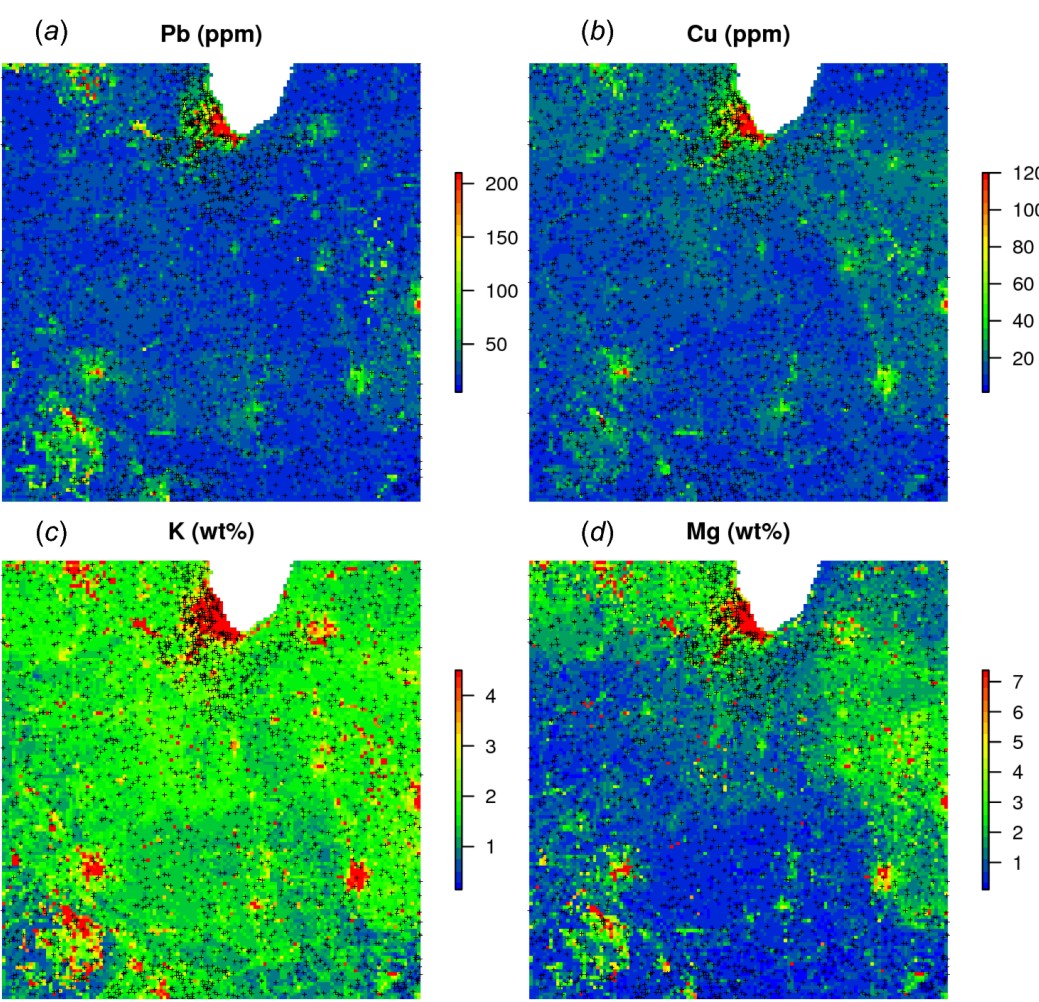

**Figure 11** **Predictions (A–D) produced for four chemical elements (wet stream sediments) from the National Geochemical Survey using a single multivariate RF model.** The study area covers the US States Illinois and Indiana. The spatial resolution of predictions is 5 km. Crosses indicate sampling locations.

```
$ NAME     : Factor w/ 233 levels "ALLENS PARK 1.5 ESE, CO US",..:
96 96 96 96 96 96 96 96 96 96 ...
$ LATITUDE: num   40.1 40.1 40.1 40.1 40.1 ...
$ LONGITUDE: num  -105 -105 -105 -105 -105 ...
$ ELEVATION: num  1567 1567 1567 1567 1567 ...
$ DATE     : Factor w/ 1462 levels "2014-11-01","2014-11-02",..: 7 13
21 35 46 67 68 69 70 75 ...
$ PRCP     : num   0 0.16 0 0 0 0.01 0.02 0.02 0.02 0.01 ...
```

```
> co_locs.sp = co_prec[!duplicated(co_prec$STATION),c("STATION",
"LATITUDE","LONGITUDE")]
> coordinates(co_locs.sp) = ~ LONGITUDE + LATITUDE
> proj4string(co_locs.sp) = CRS("+proj=longlat +datum=WGS84")
```

Even though the monitoring network consists of only 225 stations, the total number of observations exceeds 170,000. To represent '*distance*' in the time domain, we use two numeric variables—cumulative days since 1970 and Day of the Year (DOY):

```
> co_prec$cdate = floor(unclass(as.POSIXct(as.POSIXct(paste
(co_prec$DATE),
format=">
co_prec$doy = as.integer(strftime(as.POSIXct(paste(co_prec$DATE),
format="
```

variable `doy` is important to represent seasonality effects while cumulative days are important to represent long term trends. We can now prepare a spatiotemporal regression matrix by combining geographical covariates, including time and additional covariates available for the area:

```
> co_grids <- readRDS("./RF_vs_kriging/data/st_prec/boulder_grids.
rds")
> names(co_grids)

[1] "elev_1km"    "PRISM_prec"
```

where is `elev_1km` is the elevation map for the area, and `PRISM_prec` is the long-term precipitation map based on the PRISM project (http://www.prism.oregonstate.edu/normals/). Next, we also add buffer distances and bind all station and covariates data into a single matrix:

```
> co_grids <- as(co_grids, "SpatialPixelsDataFrame")
> co_locs.sp <- spTransform(co_locs.sp, co_grids@proj4string)
> sel.co <- over(co_locs.sp, co_grids[1])
> co_locs.sp <- co_locs.sp[!is.na(sel.co$elev_1km),]
> grid.distP <- GSIF::buffer.dist(co_locs.sp["STATION"], co_grids[1],
as.factor(1:nrow(co_locs.sp)))
> ov.lst <- list(co_locs.sp@data, over(co_locs.sp, grid.distP),
over(co_locs.sp, co_grids))
> ov.prec <- do.call(cbind, ov.lst)
> rm.prec <- plyr::join(co_prec, ov.prec)

Joining by: STATION

> rm.prec <- rm.prec[complete.cases(rm.prec[,c("PRCP",
"elev_1km","cdate")]),]
```

Next, we define a spatiotemporal model as:

```
> fmP <- as.formula(paste("PRCP ~ cdate + doy + elev_1km +
PRISM_prec +", dnP))
```

In other words, daily precipitation is modeled as a function of the cumulative day, day of the year, elevation, long-term annual precipitation pattern and geographical distances

to stations. Further modeling of the spatiotemporal RFsp is done the same way as with the previous 2D models:

```
> m1.prec <- ranger(fmP, rm.prec, importance = "impurity", num.trees =
150, mtry = 180)
> m1.prec

Ranger result

Type:                                Regression
Number of trees:                     150
Sample size:                         157870
Number of independent variables:     229
Mtry:                                180
Target node size:                    5
Variable importance mode:            impurity
OOB prediction error (MSE):          0.0052395
R squared (OOB):                     0.8511794

> xlP.g <- as.list(m1.prec$variable.importance)
> print(t(data.frame(xlP.g[order(unlist(xlP.g), decreasing=TRUE)
[1:10]])))

                    [,1]
cdate       93.736193
doy         87.087606
PRISM_prec   2.604196
elev_1km     2.568251
layer.145    2.029082
layer.219    1.718599
layer.195    1.531632
layer.208    1.517833
layer.88     1.510936
layer.90     1.396900
```

This shows that, distinctly, the most important covariate for predicting daily precipitation from this study area is: time i.e. cumulative and/or day of the year. The importance of cdate might not be miss-understood as a strong trend in the sense that the average amount of rainfall increases over time or the like. The covariate cdate allows the RFsp model to fit different spatial patterns for each day underpinning that the observed rainfall is different from day to day. Note that, because 1–2 covariates dominate the model, it is also important to keep mtry high (e.g., $> p/2$ where $p$ is the number of independent variables), because a standard value for mtry could result in time being systematically missed from selection.

In traditional model-based geostatistics, there are not that many worked-out examples of spatiotemporal kriging of daily precipitation data (i.e., zero-inflated variable models).
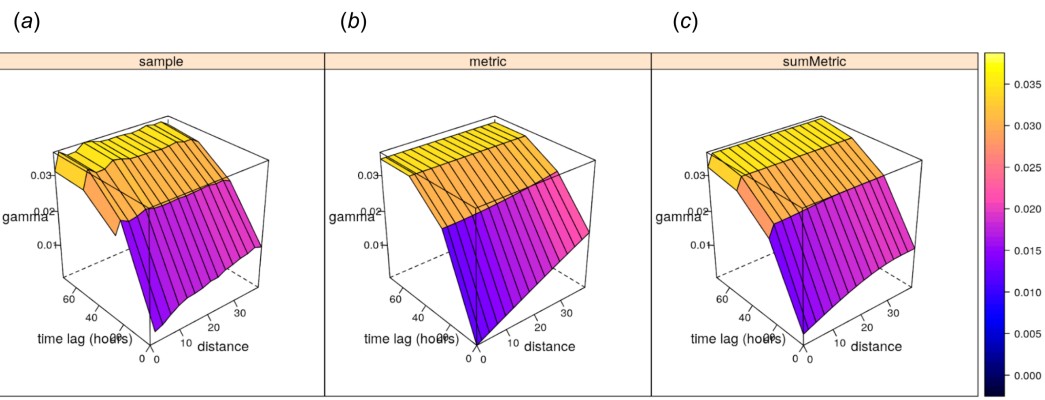

**Figure 12** Empirical, (A) and fitted metric (B, for comparison) and sum-metric (C) spatiotemporal variogram models for daily precipitation data using the spatiotemporal kriging functionality of the **gstat** package (*Gräler, Pebesma & Heuvelink, 2016*).

Geostatisticians treat daily precipitation as a censored variable (*Bárdossy & Pegram, 2013*), or cluster values e.g., (in geographical space first *Militino et al., 2015*). Initial geostatistical model testing for this data set indicates that neither of the covariates used above is linearly correlated with precipitation (with R-square close to 0), hence we use spatiotemporal ordinary kriging as a rather naïve estimator providing a geostatistical *"baseline"*. The results of fitting a spatiotemporal sum-metric model variogram using the **gstat** package functionality (*Gräler, Pebesma & Heuvelink, 2016*):

```
> empStVgm <- variogramST(PRCP~1, stsdf, tlags = 0:3)
> smmFit <- fit.StVariogram(empStVgm, vgmST("sumMetric",
+                     space=vgm(0.015, "Sph", 60, 0.01),
+                     time=vgm(0.035, "Sph", 60, 0.001),
+                     joint=vgm(0.035, "Sph", 30, 0.001),
+                     stAni=1),
+                     lower=c(0,0.01,0, 0,0.01,0, 0,0.01,0, 0.05),
+                     control=list(parscale=c(1,1e3,1,
1,1e3,1, 1,1e3,1, 1)))
```

shows the following model coefficients: (1) space —pure nugget of 0.003, (2) time — spherical model with a partial sill of 0.017, a range of 65.69 hours and a nugget of 0.007, and (3) joint —a nugget free spherical model with sill 0.009 and a range of 35 km and with spatiotemporal anisotropy of about 1 km/hour (Fig. 12).

The spatiotemporal kriging predictions can be further produced using the `krigeST` function using e.g.:

```
> predST <- krigeST(PRCP~1, stsdf[,818:833], STF(co_grids, time =
stsdf@time[823:828]),
+                   smmFit, nmax = 15, computeVar = TRUE)
```

which assumes ordinary spatiotemporal kriging model `PRCP ~ 1` with sum-metric model `smmFit` and search radius of 15 most correlated points in space and time. The cross-validation results (Leave-One-Station-Out) for RFsp approach and `krigeST` indicate that there is no significant difference between using RFsp and krigeST function: RMSE is about 0.0694 (CCC = 0.93) for `krigeST` and about 0.0696 (CCC = 0.93) for RFsp. RFsp relies on covariates such as `PRISM_prec` (PRISM-based precipitation) and `elev_1km` (elevation), so that their patterns are also visible in the predictions (Figs. 13A–13D), while `krigeST` is solely based on the observed precipitation.

Note also from Figs. 13I–13L that some hot spots in the prediction error maps for RFsp from previous days might propagate to other days, which indicates spatiotemporal connection between values in the output predictions. Even though both methods results in comparable prediction accuracy, RFsp seems to be able to reflect more closely influence of relief and impact of individual stations on predictions, and map prediction errors with higher contrast.

# DISCUSSION

## Summary results

We have defined a RFsp framework for spatial and spatiotemporal prediction of sampled variables as a data-driven modeling approach that uses three groups of covariates inside a single method:

1. geographical proximity to and composition of the sampling locations,
2. covariates describing past and current physical, chemical and biological processes,
3. spectral reflectances as direct observation of surface or sub-surface characteristics.

We have tested the RFsp framework on real data. Our tests indicate that RFsp often produces similar predictions as OK and/or RK and does so consistently, i.e., proven through repeated case studies with diverse distributions and properties of the target variable. In the case of zinc prediction for the Meuse data set, the accuracy for RFsp is somewhat smaller than for OK (Fig. 5A). In this case, RFsp with buffer distances as the only covariates evidently smoothed out predictions more distinctly than kriging. As the data size increases and as more covariate layers are added, RFsp often leads to satisfactory RMSE and ME at validation points, while showing no spatial autocorrelation in the cross-validation residuals (Figs. 5B–5C). This makes RFsp interesting as a generic predictor for spatial and spatiotemporal data, comparable to state-of-the-art geostatistical techniques already available in the packages gstat and/or geoR.

While the performance indicators show that the RFsp predictions are nearly as good as those of OK and RK, it is important to note the advantages of RFsp vs. traditional regression-kriging:

1. There is no need to define an initial variogram, nor to fit a variogram (except to check that cross-validation residuals show no spatial autocorrelation). There are no 1st and 2nd order stationarity requirements (*Goovaerts, 1997*).
2. Trend model building, which is mostly done manually for kriging, is dealt with automatically in the case of RFsp.
3. There is no need to define a search radius as in the case of kriging.

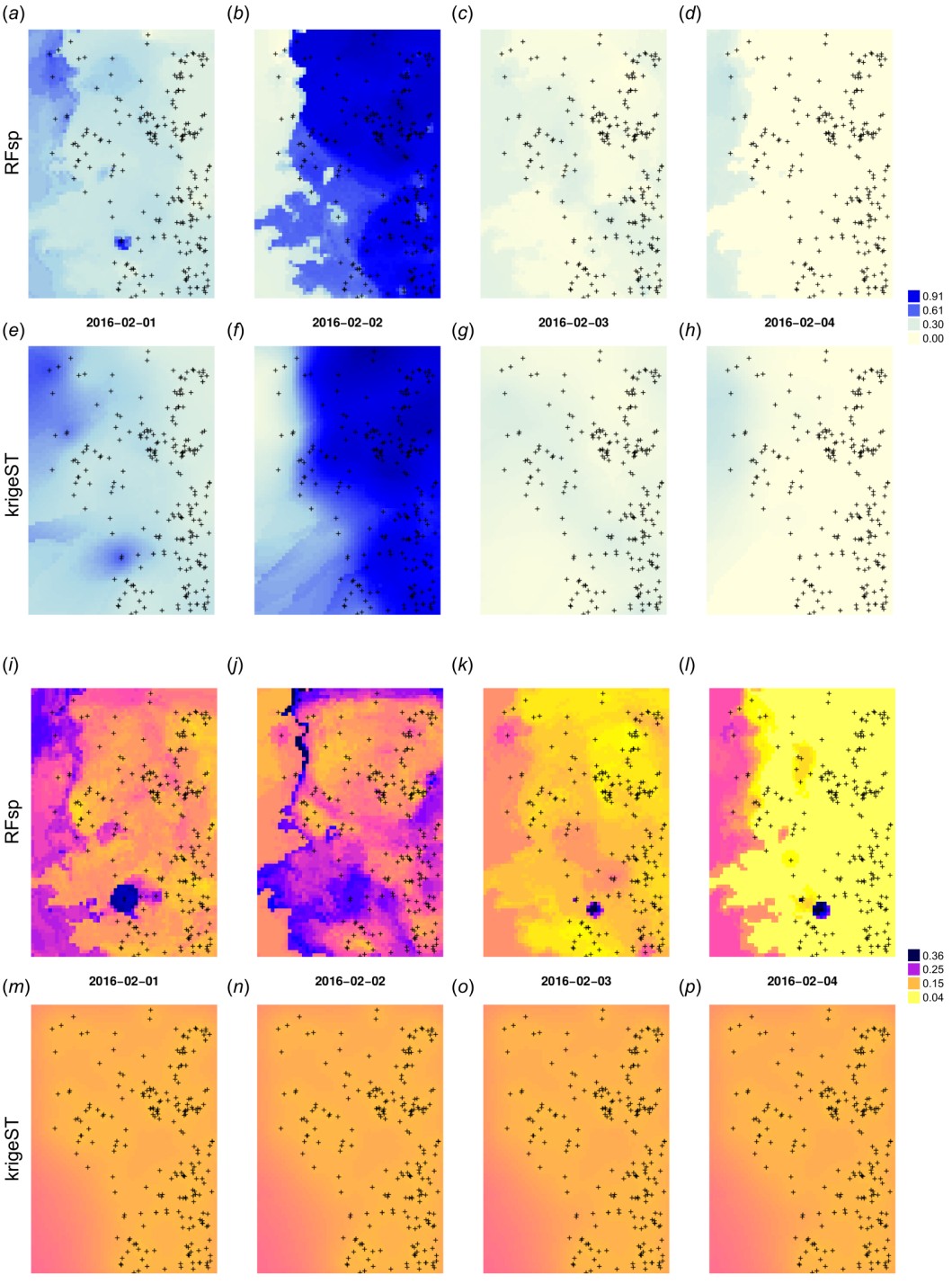

**Figure 13** Spatiotemporal predictions of daily rainfall in mm for four days in February using the RFsp and krigeST methods: RFsp predictions (A–D), krigeST predictions (E–H), standard deviation of prediction errors for RFsp (I–L), and krigeST (M–P).

4. There is no need to specify a transformation of the target variable or do any back-transformation. There is no need to deal with all interactions and non-linearities. Interactions in the covariates are dealt with naturally in a tree-based method and do not need to be manually included in the linear trend as in kriging.

5. Spatial autocorrelation and correlation with spatial environmental factors is dealt with at once (single model in comparison with RK where regression and variogram models are often fitted separately), so that also their interactions can be modeled at once.

6. Variable importance statistics show which individual observations and which covariates are most influential. Decomposition of $R^2$ as often used for linear models (*Groemping, 2006*) neglects model selection and does not straightforwardly apply to kriging.

Hence, in essence, random forest requires much less expert knowledge, which has its advantages but also disadvantages as the system can appear to be a black-box without a chance to understand whether artifacts in the output maps are result of the artifacts in input data or model limitations. Other obvious advantages of using random forests are:

- Information overlap (multicollinearity) and over-parameterization, caused by using too many covariates, is not a problem for RFsp. In the first example we used 155 covariates to model with 155 points, and this did not lead to biased estimation because RF has built-in protections against overfitting. RF can be used to fit models with large number of covariates, even more covariates than observations can be used.
- Sub-setting of covariates is mostly not necessary; in the case of model-based geostatistics, over-parameterization and/or overlap in covariates is a more serious problem as it can lead to biased predictions.
- RF is resistant to noise (*Strobl et al., 2007*).
- Geographical distances can be extended to more complex distances such as watershed distance along slope lines and or visibility indices, as indicated in the Fig. 2.

In the case of spatiotemporal data, RF seems to have ability to adjust predictions locally in space and time. Equivalent in kriging would be to use separate models for each day for example. In the precipitation case study, spatiotemporal kriging, we did not consider the issue of zero-inflation (censored variables) and have assumed a stationary field in space and time (means might vary from day to day though, but the covariance structure is the same over the entire study period). This is an obvious issue for different types of rainfall: small scale short heavy summer events, vs. widespread enduring winter precipitation, so again RFsp here shows some advantages with much less assumptions and problems with the zero-inflated nature of the data. Also, note that we could have maybe improved the spatiotemporal kriging framework with a more thorough modeling sensibly dealing with zero-inflation and the heavy skewness of the observed variable. Non-linear model based spatiotemporal statistical approaches that in general can deal with this type of random fields are e.g., models based on copulas (*Erhardt, Czado & Schepsmeier, 2015*; *Gräler, 2014*), but these are even more computational and cumbersome to implement on large datasets.

Some important drawbacks of RF, on the other hand, are:

- Predicting values beyond the range in the training data (extrapolation) is not recommended as it can lead to even poorer results than if simple linear models are used. In the way the spatiotemporal RFsp model is designed, this also applies to temporal interpolation e.g. to fill gaps in observed timeseries.
- RF will lead to biased predictions when trained with data sets that are sampled in a biased way (*Strobl et al., 2007*). To get a more realistic measure of the mapping accuracy, stricter cross-validation techniques such as the spatial declustering (*Brenning, 2012*), as implemented in the mlr package (*Bischl et al., 2016*) or similar, might be necessary.
- Size of the produced models is much larger than for linear models, hence the output objects are large.
- Models are optimized to reproduce the data of the training set, not to explain a spatial or spatiotemporal dependence structure.
- Estimating RF model parameters and predictions is computationally intensive.
- Derivation of buffer distances is computationally intensive and storage demanding.

We do not recommend using buffer distances as covariates with RFsp for a large number of training points e.g. ≫ 1,000 since the number of maps that need to be produced could blow up the production costs, and also computational complexity of such models would become cumbersome.

On the other hand, because exceptionally simple neural networks can be used to represent inherently complex ecological systems, and because computing costs are exponentially decreasing, it can be said that most of the generic Machine Learning techniques are in fact '*cheap*' and have quickly become mainstream data science methods (*Lin, Tegmark & Rolnick, 2017*). Also, we have shown that buffer distances do not have to be derived to every single observation point—for factors it turned out that deriving distances per class worked quite well. For numeric variables, values can be split into 10–15 classes (from low to high) and then again distances can be only derived to low and high values. In addition, limiting the number and complexity of trees in the random forest models (*Latinne, Debeir & Decaestecker, 2001*), e.g. , from 500 to 100 often leads to minimum losses in accuracy (*Probst & Boulesteix, 2017*), so there is certainly room for reducing size and complexity of ML models without significantly loosing on accuracy.

## Is there still need for kriging?

Given the comparison results we have shown previously, we can justifiably ask whether there is still a need for model-based geostatistics at all? Surely, fitting of spatial autocorrelation functions, i.e., variograms will remain a valuable tool, but it does appear from the examples above that RFsp is more generic and more flexible for automation of spatial predictions than any version of kriging. This does not mean that students should not bother with learning principles of kriging and geostatistics. In fact, with RFsp we need to know geostatistics more than ever, as these tools will enable us to generate more and more analyses, and hence we will also need to boost our interpretation skills. So, in short, kriging as a spatial prediction technique might be redundant, but solid knowledge of geostatistics and statistics in general is important more than ever. Also with RFsp, we still needed to fit variograms

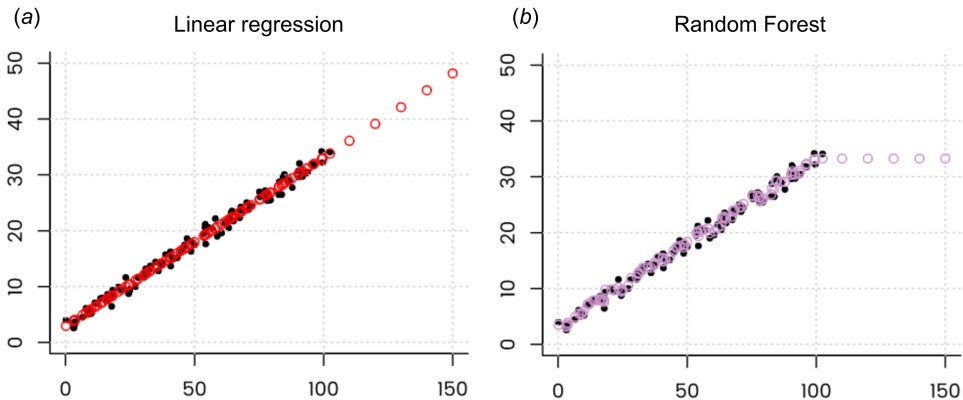

**Figure 14** **Illustration of the extrapolation problem of Random Forest.** Even though Random Forest is more generic than linear regression and can be used also to fit complex non-linear problems, it can lead to completely nonsensical predictions if applied to extrapolation domains. Image credit: Peter Ellis (http:// freerangestats.info).

for cross-validation residuals and derive occurrence probabilities etc. All this would have been impossible without understanding principles of spatial statistics, i.e., geostatistics.

While we emphasize that data-driven approaches such as RF are flexible and relatively easy to use because they need not go through a cumbersome procedure of defining and calibrating a valid geostatistical model, we should also acknowledge the limitations of data-driven approaches. Because there is no model one can also not inspect and interpret the calibrated model. Parameter estimation becomes essentially a heuristic procedure that cannot be optimized, other than through cross-validation. Finally, extrapolation with data-driven methods is more risky than with model-based approaches. In fact, in the case of RF, extrapolation is often not recommended at all—all decision-tree based methods such as RFs or Boosted Regression Trees can complete fail in predictions if applied in regions that have not been used for training (Fig. 14B).

### Are geographic covariates needed at all?

The algorithm that is based on deriving buffer distance maps from observation points is not only computationally intensive, it also results in a large number of maps. One can easily imagine that this approach would not be ready for operational use where ≫ 1,000 as the resources needed to do any analysis with such data would easily exceed standard budgets. But are buffer distances needed at all? Can the geographical location and proximity of points be included in the modeling using something less computationally intensive?

*McBratney, Santos & Minasny (2003)* have, for example, conceptualized the so-called "*scorpan*" model in which soil property is modeled as a function of:

- (auxiliary) **s**oil properties,
- **c**limate,
- **o**rganisms, vegetation or fauna or human activity,
- **r**elief,
- **p**arent material,

- **a**ge i.e. the time factor,
- **n** space, spatial position,

It appears that also **s** and **n** could be represented as a function of other environmental gradients. In fact, it can be easily shown that, as long as there are enough unique covariates available that explain the majority of physical and chemical processes (past and current) and enough remote sensing data that provides spectral information about the object/feature, each point on the Globe can be defined with an unique *'signature'*, so that there is probably no need for including spatial location in the predictive mapping at all.

In other words, as long as we are able to prepare, for example, hundreds of covariates that explain in detail uniqueness of each location (or as long as an algorithm can not find many duplicate locations with unique signature), and as long as there are enough training point to describe spatial relations, there is probably no need to derive buffer distances to all points at all. In the example by *Ramcharan et al. (2018)*, almost 400,000 points and over 300 covariates are used for training a MLA-based prediction system: strikingly the predicted maps show kriging-like pattern with spatial proximity to points included, even though no buffer distances were ever derived and used. It appears that any tree-based machine learning system that can *'learn'* about the uniqueness of a geographical location will eventually be able to represent geographical proximity also in the predictions. What might be still useful is to select a smaller subset of points where hot-spots or points with high CV error appear, then derive buffer distances only to those points and add them to the bulk of covariates.

*Behrens et al. (2018a)* have recently discovered that, for example, DEM derivatives correlate derived at coarser scales correlate more with some targeted soil properties than the derivatives derived as fine scales; in this case, scale was represented through various DEM aggregation levels and filter sizes. Some physical and chemical processes of soil formation or vegetation distribution might not be visible at finer aggregation levels, but then become very visible at coarser aggregation levels. In fact, it seems that spatial dependencies and interactions of the covariates can be explained simply by aggregating DEM and the derivatives. For a long time physical geographers have imagined that climate, vegetation and similar are non-linear function of longitude and latitude; now it appears that the inverse of this could be also be valid.

## Remaining methodological problems and future directions

Even though MLA has proven to be efficient in boosting spatial prediction performance, there still remain several methodological problems before it can be widely applied, for example:

- How to generate spatial simulations that accurately represents spatial autocorrelation structure using RF models?
- How to produce predictions from and at various block support sizes—from point support data to block support data and vice versa?
- How to deal with extrapolation problems (both in feature and geographical spaces)?
- How to account for spatial and spatiotemporal clustering of points?

Although Machine Learning is often very successful in spatial prediction, we should not be over-relaxed by its flexibility and efficiency of crunching data. Any purely data or pattern driven classifier or regressor is a rather mechanical approch to problem solving. It ignores all of our knowledge of processes and relationships that have been documented and proven to work over and over. It does not have an explicit (geo)statistical model as a starting point, so that no mathematical derivations are possible at all. Also, just adding more and more data to the system does not necessarily mean that the predictions will automatically become better (*Zhu et al., 2012*). The main difficulty ML user experience today is to explain how a particular algorithm has come to its conclusions (*Hutson, 2018*). One extreme projection of blind over-use of ML and A.I. is that it could leave us with less and less capacity to generate knowledge. In that context, what maybe could seem as a logical development direction for Machine Learning is development of hybrid use of data and models, i.e., an A.I. systems that not only mechanically mines data, but also mines models and knowledge and extends from testing accuracy improvements to testing more complex measures of modeling success such as model simplicity, importance of models across various domains of science even testing of mathematical proofs (*Lake et al., 2017*). Such systems would have been at the order of magnitude more complex than Machine Learning, but, given the exponential growth of the field of A.I., this might not take decades to achieve.

### One model to rule them all?

Given that with RF multiple variables can be predicted at once, and given that all global data from some theme such as soil science, meteorology etc. could be put into a single harmonized and integrated database, one could argue that, in the near future, a single machine learning model could be fitted to explain all spatial and/or spatiotemporal patterns within some domain of science such as soil science, meteorology, biodiversity etc. This is assuming that ALL observations and measurements within that domain have been integrated and pre-processed/harmonized for use. Such models could potentially be used as *'knowledge engines'* for various scientific fields, and could be served on-demand, i.e., they would generate predictions only if the predictions are required by the users.

These data set and models would be increasingly large. In fact, they would probably require super computing power to update them and efficient data storage facilities to serve them, hence the current state-of-the-art data science might gradually move from managing Big Data only, to managing Big Data and Big Models.

## CONCLUSIONS

We have shown that random forest can be used to generate unbiased spatial predictions and model and map uncertainty. Through several standard textbook datasets, we have shown that the predictions produced using RFsp are often equally accurate (based on repeated cross-validation) than equivalent linear geostatistical models. The advantages of random forest vs. linear geostatistical modeling and techniques such as kriging, however, lies in the fact that no stationarity assumptions need to be followed, nor is there a need to specify transformation or anisotropy parameters (or to fit variograms at all).

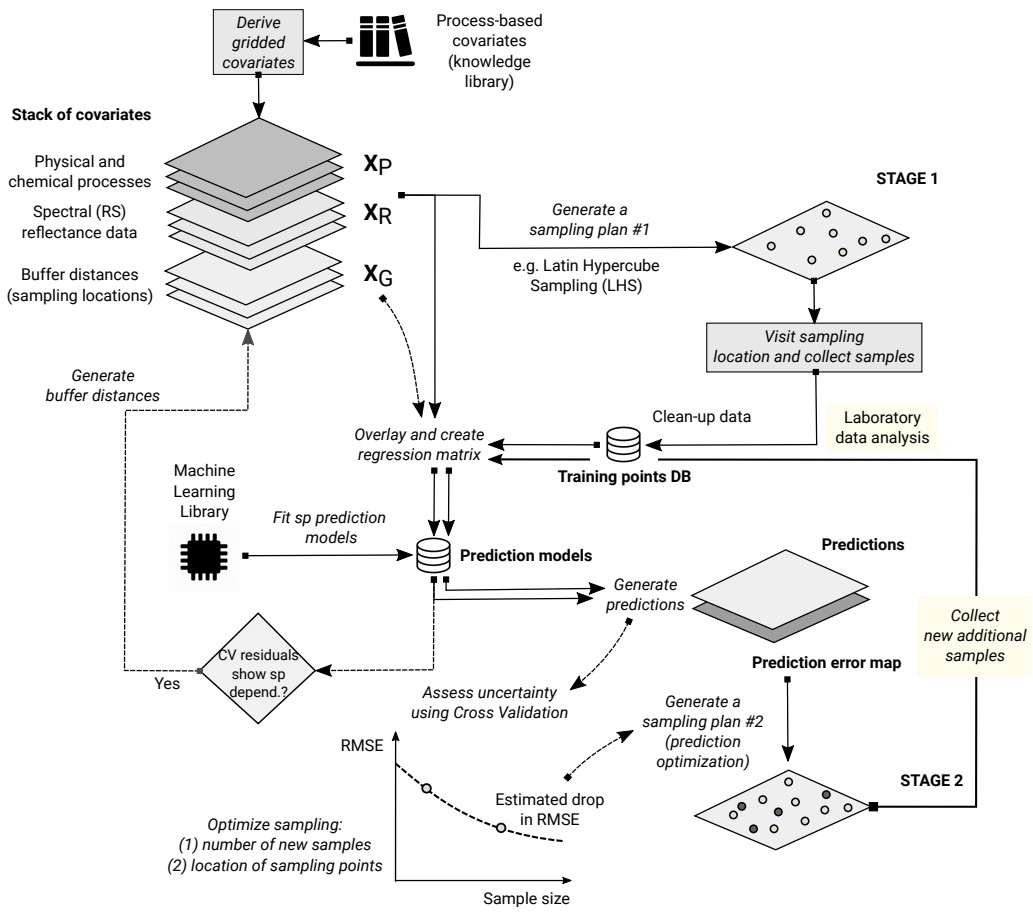

**Figure 15 The recommended two-stage accuracy-driven framework for optimizing spatial predictions based on RFsp (see also Eq. (18)).** In the first stage, minimum number of objectively sampled points are used to get an initial estimate of the model. In the second stage, the exact number of samples and sampling locations are allocated using the prediction error map, so that the mapping accuracy can be brought towards the desired or target confidence intervals.

This makes RF fairly attractive for automated mapping applications, especially where the point sampling is representative (extrapolation minimized) and where relationship between the target variable, covariates and spatial dependence structure is complex, non-linear and requires localized solutions. Some serious disadvantage of using RFsp, on the other hand, is sensitivity to input data quality and extrapolation problems (Fig. 14). The key to the success of the RFsp framework might be the training data quality—especially quality of spatial sampling (to minimize extrapolation problems and any type of bias in data), and quality of model validation (to ensure that accuracy is not effected by overfitting).

Based on the discussion above, we can recommend a two-stage framework explained in Fig. 15, as possibly the shortest path to generating maximum mapping accuracy using RFsp whilst saving the production costs. In the first stage, initial samples are used to get an estimate of the model parameters, this initial information is then used to optimize predictions (the second stage) so that the mapping objectives can be achieved with

minimum additional investments. The framework in Fig. 15, however, assumes that there are (just) enough objectively sampled initial samples, that the RF error map is reliable, i.e., accurate, that robust cross-validation is used and a reliable RMSE decay function. Simple decay functions could be further extended to include also objective *'cooling'* functions as used for example in *Brus & Heuvelink (2007)*, although these could likely increase computational intensity. Two-stage sampling is already quite known in literature (*Hsiao, Juang & Lee, 2000*; *Meerschman, Cockx & Van Meirvenne, 2011*; *Knotters & Brus, 2013*), and further optimization and automation of two-stage sampling would possibly be quite interesting for operational mapping.

Even though we have provided comprehensive guidelines on how to implement RF for various predictive mapping problems—from continuous to factor-type variables and from purely spatial to spatiotemporal problems with multiple covariates—there are also still many methodological challenges, such as derivation of spatial simulations, derivation of buffer distances for large point data sets, reduction of extrapolation problems etc, to be solved before RFsp can become fully operational for predictive mapping. Until then, some traditional geostatistical techniques might still remain preferable.

## ACKNOWLEDGEMENTS

We are grateful to the developers of the original random forest algorithms for releasing their code in the Open Source domain (*Breiman, 2001*), Philipp Probst for developing algorithms for fine-tuning of RF and implementing the Quantile Regression Forests, and the developers of the spatial analysis packages GDAL, rgdal, raster, sp (*Pebesma, 2004*; *Bivand et al., 2008*), and SAGA GIS (*Conrad et al., 2015*), on top of which we have built work-flows and examples of applications.

### Funding

The contributions by Benedikt Gräler have been funded by the German Federal Ministry for Economic Affairs and Energy under grant agreement number 50EE1715C. The funders had no role in study design, data collection and analysis, decision to publish, or preparation of the manuscript.

### Grant Disclosures

The following grant information was disclosed by the authors:
German Federal Ministry for Economic Affairs and Energy: 50EE1715C.

### Competing Interests

Tomislav Hengl is employed by the Envirometrix Ltd., Wageningen, Gelderland, Netherlands (http://envirometrix.net). Marvin N. Wright is employed by the Leibniz Institute for Prevention Research and Epidemiology –BIPS, Bremen (https://www.bips-institut.de/en/the-institute/departments/biometry-and-data-management/statistical-methods-in-genetics-and-life-course-epidemiology.html).

## Author Contributions

- Tomislav Hengl and Madlene Nussbaum conceived and designed the experiments, performed the experiments, analyzed the data, contributed reagents/materials/analysis tools, prepared figures and/or tables, authored or reviewed drafts of the paper, approved the final draft.
- Marvin N. Wright performed the experiments, analyzed the data, contributed reagents/materials/analysis tools, prepared figures and/or tables, approved the final draft.
- Gerard B.M. Heuvelink analyzed the data, authored or reviewed drafts of the paper, approved the final draft, mathematical syntax checking.
- Benedikt Gräler analyzed the data, contributed reagents/materials/analysis tools, prepared figures and/or tables, approved the final draft, spacetime kriging.

## Data Availability

All code used in the paper is also available at: https://github.com/thengl/GeoMLA.

Ranger package for R is available under the GPL at: https://github.com/imbs-hl/ranger.

## Supplemental Information

Supplemental information for this article can be found online at http://dx.doi.org/10.7717/peerj.5518#supplemental-information.

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
