# Peer review of "Random forest as a generic framework for predictive modeling of spatial and spatio-temporal variables"

_PeerJ, doi:10.7717/peerj.5518_

## Round 0.1 · original submission · Major Revisions

Thank you for submitting your paper "Random Forest as a generic framework for predictive modeling of spatial and spatio-temporal variables" to PeerJ. Your ms has now been reviewed by two experts in the field and I enclose their responses to your work. Both the reviewers and I found the ms potentially contributing to the readers of PeerJ, however, the reviewers have also provided extensive and thoughtful comments that should be fully and carefully answered before further consideration for publication. Please revise the ms in response to the reviewer's comments and resubmit it to the journal website, so that it can be given further consideration. The ms and your letter of response will be reviewed again and the final decision is yet unknown.

Reviewer 1 ·

Basic reporting

1.1 Clear, unambiguous, professional English language used throughout.

Yes, just a few colloquial expressions, e.g. a method "blows up".

Abstract: "lower number of" -> "fewer" points...

1.2 Intro & background to show context. Literature well referenced & relevant.

Yes

1.3 Structure conforms to PeerJ standards, discipline norm, or improved for clarity.

No. The Discussion goes much further than the evidence in the Results, it is largely speculative without support from the Results.

1.4 Figures are relevant, high quality, well labelled & described.

Mostly.

1.5 Raw data supplied (see PeerJ policy).

Yes

Experimental design

2.1 Original primary research within Scope of the journal.

Yes

2.2 Research question well defined, relevant & meaningful. It is stated how the research fills an identified knowledge gap.

Yes. The authors propose a spatial prediction framework based on random forests, including distances to observation points, as a machine learning alternative to model-based geostatistics. They give cogent reasons why this might be desireable.

2.3 Rigorous investigation performed to a high technical & ethical standard.

Not completely. The paper wanders through several examples without a clear formulation of the RPsp model, just a description in words. The examples are diverse and illustrate various techniques without sufficient justification or motivation. This reader had the impression that the paper was pasted together from the tutorial (supplementary material). The later sections of the paper are speculation unsupported by what is presented in the paper. These sections read as the subjective opinions of the authors, and although they are interesting and provocative, they are not supported by any tests in this paper.

In the case studies, difference maps for both predictions and standard errors would add to the reader's understanding of the differences, and could then be discussed.

2.4 Methods described with sufficient detail & information to replicate.

Not completely. The code is missing key parts (e.g., loading of required packages). By digging into the supplementary material at github (linked) it is possible to find the missing pieces, but the presentation in the paper is not self-contained.

Validity of the findings

3.1 Impact and novelty not assessed. Negative/inconclusive results accepted. Meaningful replication encouraged where rationale & benefit to literature is clearly stated.

This paper could in fact have a large impact, as it provides a machine learning approach to spatial prediction.

3.2 Data is robust, statistically sound, & controlled

All data is from standard datasets used in geostatistical examples, presumably controlled by their authors, not these.

3.3 Conclusions are well stated, linked to original research question & limited to supporting results.

No. See comments under 2.2 above and details below.

3.4 Speculation is welcome, but should be identified as such.

Too much speculation, too far from what is actually done.

Additional comments

The paper has many holes and inconsistencies. The following is a list in line order of some of the issues that should be addressed.

Abstract L31: "the high sensitivity of predictions to input data quality" -- really? how shown?

L33 "close-to-linear relationships" in the covariates?

L53 mention other R packages for RF, e.g. rpart, probably the most commonly used

L60 linear features -> *linear boundaries in the resulting map*

L78 i.e. -> e.g.? If restricted to 2D you should state earlier or even in title. In fact you have a 3D example (NRCS).

L80 spatiotemporal phenomena -> spatio-temporally distributed variables?

L109 omit "basically", adds nothing

Do we need so much background on MLR & RK? nothing wrong with it, but maybe for a text, not a paper?

L149 account for the *spatial correlation* or other non-independence of the *residuals*. Mention REML (Lark)

L160 i.e. -> e.g., can be anything; also not necessary here

L230ff, eqn (19) out of nowhere of a sudden remote sensing is included. This is not inherent in the RFsp idea, but it is a useful decomposition for situations where RS provides useful covariates. Conceptually X_R fit into the RF model in the same way as X_P -- there is no mathematical difference. If RS is central to the idea, it should be in the title. But it seems this just limits the approach unnecessarily.

L235ff "Assuming that the RFsp is fitted only using the XG, the predictions will likely look similar to OK. If all covariates are used (Eq.19), RFsp will likely produce similar results as regression-kriging" -- this is to be determined by the study, it has not been shown at all to this point. Perhaps rephrase "since the information content is the same as... the results may resemble..."

L247 remove "etc." Add dIstance to features such as water bodies, cities ...

L251 Yes visibility distances can be calculated, but what theory predicts that these may be useful?

Of this list, which should be more useful? You already rejected (1) above (just use coords) because of linear artefacts, why include it here? One reason to use it would be the analogy to UK using trend surfaces -- if there really is a trend (especially if non-linear or with a threshold) these could be useful. And -- Figure 4 below (and others) show this! So the statement that you only use local buffers is not true. You also use coordinates and then compare the two approaches.

Here we need a mathematical discription of the buffer distances approach. We only see it in action in the examples, but never with any formula. Yes this is simple but needs to be explicit in mathematical form. It is not obvious from the text that the RF is built with all the buffer distances to all the points -- yes, this can be inferred from the code, but it is a major point of the approach, which should be brought out explicitly. Also, the distinction between using distances to all points, and classes of buffer distances, should be explained here.This is used later in the paper with almost no explanation. It's unclear (except by digging into the code at github for that example) how both approaches are implemented.

L264 vague and not supported. "easily"? how? suggest to omit. Concentrate on the value of using distance to points.

L267 X-validation RMSE, but model R^2 does not quantify predictive accuracy, only fitting accuracy. Or do you mean CV R^2?

L270 Spatial bias or residual uncertainty? See Fig. 3(d) caption, seems correct there.

L272 z-scores of what? Fig. 3(c) uses terms "nominal" and "coverage" probability w/o explanation

L293ff. It would be useful to also compare to a classical geostatistical estimate using e.g., gstat or similar

L296 We need some explanation of this dataset -- what are its characteristics, therefore what approaches are indicated? The important fact is that we think the metal contents are related to river flooding carrying upstream sediments.

L299 ff. the explanations from geoR functions are not needed. "shows that..." just state as a fact (backed up by the references, which are already given)

FIgure 4 also shows using coordinates, but you said above it was (1) not desireable (2) you wouldn't do it. Also you do not show an observed-predicted plot for that case, so why show the map?

Figure 4 "and correlation plots based on the 5–fold cross-validation for OK and RFsp (last row, solid line: lowess scatterplot smoother)."?? nothing there

L332 "we may add global surface water occurrence (Pekel et al., 2016) and the LiDAR-based digital elevation model (DEM, http://ahn.nl) as potential covariates explaining zinc concentration" This is unmotivated. Why could these be good predictors? In this area elevation may be related to flooding, but the authors do not explain this. How could surface water occurrence explain Zn?

L334 function spc() is not explained -- which package? and more importantly, why used? It is not in any of the loaded libraries (sp, plotKML, geoR, rgdal). Also, some used packages are not explicitly loaded: GSIF, and even ranger!! With some digging I found it in GSIF. But its function is not at all explained in the paper, it appears as if by magic.

L347-355 largely repeats the advantages proposed for RFsp earlier in the paper, this is not specific to the case study

L369 This information of likfit() should be given when the function is first used.

L434 carson1km <- readRDS("data/NRCS/carson_covs1km.rds") where does this RData file come from? Also the .csv.

L436 In what sense is this a "pragmatic" approach? It seems based on actual lab. precisions.

This section is interesting but adds a complexity that is not reflected in the paper title nor objectives.

L448ff (Geochemical survey) There is a lot of R manipulation here, all very useful, but it reads as a tutorial, not as a motivated procedure. This whole section seems a really wrong way to go about the objective. Yes it can be done mathematically, but you just put all the predictands together, to later disaggregate them with the indicator predictors. I fail to see the advantage vs. fitting separate models.

L489ff (Daily precip Boulder) is an interesting application, but it should be compared with spatio-temporal kriging. L514 "This shows that RFsp connects space and time in a similar way as the model-based geostatistics.", no you didn't show that, what is the "same way as" supposed to mean?

L511 "The single spatiotemporal model can now be used to predict anywhere within the spacetime domain" If you mean only within the space-time region of calibration, yes; but if you mean in general, no, because RF can not predict outside its range of calibration (e.g., daily precip more than any that was used to calibrate the model). Please clarify. L552 mentions this point; but "not recommended" is too weak, it should be "impossible". L590 "serious extrapolation" should be "any extrapolation". What would "serious" be, anyway?

L532-536 repeats previous points, this is the third mention, not including the abstract.

L538 "save" here means what?

L563 "exceptionally simple neural networks can be used to represent inherently complex ecological systems" proof? references?

L566 "Also, we have shown that buffer distances do not have to be derived to every single observation point — for factors it turned out that deriving distances per class worked quite well."

Not true for the first model: "layer.* are buffer distances to each individual point"; but then suddenly L414 "in this case buffer distances are derived to each class" -- why now and not for the first case? And why not do this in general? How can we decide between distances to points or classified distances?

Also, you did not test this: " For numeric variables, values can be split into 10–15 classes (from low to high) and then again distances can be only derived to low and high values." This could be true but the paper does not show this with any comparative test.

L569ff "Limiting the number and complexity of trees in the random forest models..." Again, you did not test this, so how can you state it for your RFsp approach?

L582 "Also with RFsp, we still needed to fit variograms for cross-validation residuals and derive occurrence probabilities etc." ??? I did not see this. See also the comment for L635.

L592ff "Are geographic covariates needed at all?" This whole section is speculation and unsupported by the studies in the Results.

L635 Where do these simulations come in? Why do you need to simulate? You have the data points and will use them for the RFsp model, what remains to be simulated? Why do you need to know the spatial structure? You have said throughout you can just use the distance buffers with no need to know the structure, as in variogram modelling. See also the comment for L582.

L662ff "One model to rule them all" Pure speculation, suitable maybe for a session in the pub but completely unsupported here.

L690ff This two-stage framework and the accompanying diagram are interesting, but again unsupported by the results of this study.

L702ff. It seems you have solved all these problems in your paper, even the "large point data sets" (by using class distance buffers).

L707 Acknowledgements: These were not direct helps to your study, they were previous work which you used. It is not common to acknowledge them, unless their authors gave you direct aid.

Reviewer 2 ·

Basic reporting

See below

Experimental design

See below

Validity of the findings

See below

Additional comments

General comments:
In this paper the authors introduce a random forest for spatial predictions framework (RFsp) where geographical proximity effects are incorporated into the prediction by using buffer distances from observation points as explanatory variables. The approach is illustrated using diverse spatial and spatio-temporal datasets, and results are compared to kriging. The paper is clearly written and the approach is interesting and seems technically sound, with the pros and cons of each approach clearly outlined. A few issues still need to be addressed before the paper can be published:
1. The authors seem to imply that regression kriging, universal kriging and kriging with an external drift (KED) are the same methods and give similar predictions (lines 140-144). This statement is not correct as KED fits a local regression model within each search window while regression kriging is based on a unique regression model.
2. The critical step of back-transforming results needs to be better explained. In particular, it is not clear how results can be back-transformed for a Box-Cox transformation with η≠0. For the case of a lognormal transform, Eqs. (13) and (14) are only correct for simple kriging as the Lagrange multiplier is missing. Last, how do the authors compute the kriging variance used to back-transform RK estimates (Line 155)? Do they combine the residual kriging variance with the uncertainty attached to regression estimates?
3. At several locations across the manuscript, the authors mentioned that some implementations of the algorithms are computationally intensive. It would be useful for the reader to provide some estimates of CPU time.
4. The space-time example lacks a comparison with space-time kriging.
5. In any cross-validation of datasets with repeated samples (e.g., time series) one should remove the entire set of co-located observations when estimating a particular location to avoid the under-estimation of prediction errors mentioned on line 643.

Detailed comments:
1. Line 79. The term “raster map” might be more appropriate than “gridded map”.
2. Line 168. RK does not assume a linear relationship between dependent and independent variables as any type of regression model can be fitted.
3. Line 240. Replace “froms the basis” by “forms the basis”.
4. Line 258. Write “resistance distances” instead of “resistence distances”
5. Figure 6. The caption should be clarified, e.g. by adding lettering to refer to the different figure panels.
6. Lines 378-379. Could the homogeneity of the UK prediction error map (Fig. 7) be caused by the use of the same color scheme as for the RK prediction error map?
7. Fig.11. Why is the shape of the map a rectangle unlike the actual shape of the States of Indiana and Illinois. Does the method impose such constraint? This should be clarified.
8. Fig. 12. All the maps seem to display a discontinuity (horizontal line), this should be discussed.
9. Line 614. Write “as long as”.
10. Line 630-631. This sentence is confusing and needs to be completely rewritten.

---

## Round 0.2 · Minor Revisions

There is still a problem with files stored in the github. Without knowing exactly where the file is, you have to poke around the directory structure to search for it (the file is levels below the project root directory, and the root does not include a "Data" directory - you have to drill down a level to find anything that suggests you are on the right track). In the interests of reproducibility, the data used should be explicitly provided (via a URL is fine) rather than only being available to the informed R users who know enough to go and find it. The manuscript says "A complete benchmarking of the prediction efficiency is documented in R code and can be obtained via the GitHub repository at https://github.com/thengl/GeoMLA. All datasets used in this paper are either part of an existing R package or can be obtained from the GitHub repository." but it would certainly be a good idea for the authors to say in the manuscript that the data are under the /RF_vs_kriging/data directory. As for the directions to look at the tutorial on Github, that is not bad advice, but it could be mentioned in the text at strategic places.

Also, the relatively minor comment of Reviewer 2 needs to be addressed.

Reviewer 1 ·

Basic reporting

I choose not to evaluate the 2nd version for reasons which I have communicated to the editor.

Experimental design

I choose not to evaluate the 2nd version for reasons which I have communicated to the editor.

Validity of the findings

I choose not to evaluate the 2nd version for reasons which I have communicated to the editor.

Additional comments

I choose not to evaluate the 2nd version for reasons which I have communicated to the editor. Having seen how the authors responded to my concerns about the original preprint, further comment would obviously be pointless and only lead to another argumentative response.

Reviewer 2 ·

Basic reporting

Good

Experimental design

Good

Validity of the findings

Good

Additional comments

The authors have addressed most of the comments I raised in my initial review. A good reference that explains that KED conducts a local fitting of the regression model within each search window are the books by Hans Wackernagel (1995) and Pierre Goovaerts (1997).

---

## Round 0.3 · accepted · Accept

The paper can be accepted now.